# Non-monotonic pressure dependence of high-field nematicity and magnetism in CeRhIn$_5$

Toni Helm [1,2✉], Audrey D. Grockowiak[3], Fedor F. Balakirev [4], John Singleton [4], Jonathan B. Betts[4], Kent R. Shirer[1], Markus König[1], Tobias Förster[2], Eric D. Bauer [5], Filip Ronning[5], Stanley W. Tozer[3] & Philip J. W. Moll [1,6✉]

CeRhIn$_5$ provides a textbook example of quantum criticality in a heavy fermion system: Pressure suppresses local-moment antiferromagnetic (AFM) order and induces superconductivity in a dome around the associated quantum critical point (QCP) near $p_c \approx 23$ kbar. Strong magnetic fields also suppress the AFM order at a field-induced QCP at $B_c \approx 50$ T. In its vicinity, a nematic phase at $B^* \approx 28$ T characterized by a large in-plane resistivity anisotropy emerges. Here, we directly investigate the interrelation between these phenomena via magnetoresistivity measurements under high pressure. As pressure increases, the nematic transition shifts to higher fields, until it vanishes just below $p_c$. While pressure suppresses magnetic order in zero field as $p_c$ is approached, we find magnetism to strengthen under strong magnetic fields due to suppression of the Kondo effect. We reveal a strongly non-mean-field-like phase diagram, much richer than the common local-moment description of CeRhIn$_5$ would suggest.

[1] Max Planck Institute for Chemical Physics of Solids, 01187 Dresden, Germany. [2] Dresden High Magnetic Field Laboratory (HLD-EMFL), Helmholtz-Zentrum Dresden-Rossendorf, 01328 Dresden, Germany. [3] National High Magnetic Field Laboratory, Tallahassee, FL 32310, USA. [4] National High Magnetic Field Laboratory Pulsed-Field Facility, MS-E536, Los Alamos National Laboratory, Los Alamos, NM 87545, USA. [5] Los Alamos National Laboratory, Los Alamos, NM 87545, USA. [6] Laboratory of Quantum Materials (QMAT), Institute of Materials (IMX), École Polytechnique Fédérale de Lausanne, 1015 Lausanne, Switzerland. ✉email: t.helm@hzdr.de; philip.moll@epfl.ch

The physical properties of cerium-based heavy-fermion superconductors are strongly governed by the Ce $4f^1$ electrons and their interaction with the itinerant charge carriers[1,2]. Due to the small scale of the relevant energies associated with the hybridization of the $4f$-electrons with the conduction bands, small changes in the chemical composition, magnetic field, or pressure strongly influence the ground state and frequently lead to quantum critical phenomena[3–7]. Here we focus on CeRhIn$_5$, a local-moment antiferromagnet characterized by a Néel temperature of $T_N = 3.85$ K[8]. The antiferromagnetic (AFM) order is suppressed under pressure, and superconductivity emerges in a dome located around the associated AFM quantum critical point (QCP) at $p_c \approx 23$ kbar[9]. The picture of pressure-induced quantum criticality is further supported by the observation of non-Fermi-liquid behavior in its vicinity[9–11]. Recent experiments have provided evidence for a second QCP at ambient pressure and under strong magnetic fields, $B_c(p = 0) \approx 50$ T[12,13]. The temperature dependence of the resistivity at the field-induced QCP is well described by a similar power law, $\rho(B = 50\,\text{T}, p = 0) \propto T^{0.91}$, compared to the behavior at the zero-field QCP, $\rho(B = 0, p = 23\,\text{kbar}) \propto T^{0.85}$[9,13]. Interestingly, the magnitude of $B_c$ is almost completely independent of the field orientation, despite the presence of magnetic anisotropy $\chi_c/\chi_a \approx 2$[14]. This isotropy of $B_c$ provides a first hint that CeRhIn$_5$ in high magnetic fields goes beyond a simplistic picture of a local-moment magnet.

Recently, a field-induced phase transition was observed at intermediate magnetic fields $B^* \geq 28$ T[12,15,16], and a nematic character of this high-field phase has been reported, based on the sudden emergence of an in-plane resistivity anisotropy[13]. Magnetic probes, such as magnetization and torque, however, show hardly any features in the relevant field-temperature range[13,14], thus rendering a metamagnetic origin of the $B^*$ transition unlikely. A nematic state, as proposed, electronically breaks the $C4$ rotational symmetry in the $(a, b)$ plane, and hence must be reflected by a small lattice distortion[17], which has recently been verified by magnetostriction experiments confirming the thermodynamic character of the transition at $B^*$[18]. For these reasons we shall refer to the sudden and strong field-induced transport anisotropy at $B^*$ as nematic for simplicity. Yet its microscopic origin remains a highly active area of research, and the nematic picture is constantly expanded, refined as well as challenged. The main open questions concern the explicitly symmetry-breaking role of the in-plane magnetic field[19], for example, through a modification of the crystal-electric-field schemes, as well as potential changes of the microscopic magnetic ordering that might remain undetected by measurements of the averaged magnetization and torque. Here, the recent breakthroughs in pulsed magnetic field neutron scattering[20] and other microscopic techniques would be most insightful. One of the main results of the present work is to trace the field scale, $B^*(p)$, into the high-pressure regime regardless of its origin.

CeRhIn$_5$ presents a unique opportunity to tune a highly correlated system between a diverse set of ground states, including unconventional superconductivity, local-moment magnetism, a putative (spin- or charge-) nematic, and a heavy Fermi liquid[8,13,21,22]. Here we chart the unknown territory of combined high field/high pressure, to serve as a testbed for theoretical approaches tackling this problem. Most importantly, the entire phase diagram can be mapped within one clean single crystal, hence avoiding the complexities of sample dependence and non-stoichiometry, by the use of the two least invasive tuning parameters, pressure and magnetic field. While conceptually appealing, this presents a formidable experimental challenge: transport experiments on metallic samples in pulsed magnetic fields need to be combined with diamond-anvil pressure cells (DACs) and $^3$He temperatures. Inspired by previous experiments in high fields and

high pressures[23–26], we here present a new approach combining focused ion beam (FIB) crystal micromachining[27] and DACs[26] made from plastic for multi-terminal measurements of transport anisotropy (SEM images of the devices can be found in Fig. 1a and Supplementary Fig. 1). To minimize heating due to eddy currents in pulsed magnetic fields, the body of the pressure cell and the $^3$He cryostat was made entirely from plastic (for further details see "Methods").

In the following, we detail three main experimental observations uncovered by this study: first, the nematic onset field $B^*$, characterized by an anisotropy jump and a first-order-like hysteretic behavior, grows with applied pressure from 28 T at ambient pressure to around 40 T for close to $p^* \approx 20$ kbar (see Fig. 2 left panel). At the same time the hysteretic transition, hallmark of entrance into the nematic state, continuously diminishes until $p^*$, at which it vanishes completely. Second, the critical field for the AFM transition, $B_c$, also shifts to higher fields upon pressure increase—exceeding 60 T at $p \approx 17$ kbar, in contrast to the zero-field suppression of the AFM order around this pressure. Our third observation suggests a field-induced reentrance of AFM at pressures above $p_c$. The critical field, $B_c$, and the critical pressure, $p_c$, both terminate the symmetry-breaking AFM dome and therefore must be connected by a continuous line of phase transitions. Indeed, we do observe a discontinuity in the slope of the magnetoresistivity (MR) above $p_c$, tracing out an upward line $B_{c,low}$ that increases with increasing pressure. It is natural to associate this field with the field-induced reentrance of AFM order as expected.

## Results

### Multi-terminal magnetoresistance measurements in a diamond-anvil pressure cell

The electric resistivity is a formidable indicator for any changes in a metal, yet as a non-thermodynamic probe it must be complemented by other measurements to firmly establish the nature and symmetry of the phases occurring in a sample. The challenging experimental environments under investigation here, however, preclude such complementary measurements at present. We thus adopt the following strategy: first, the samples are characterized in the low-pressure/low-field range, in which thermodynamic probes have well established the phase diagram. This allows us to robustly identify the resistive signatures that occur at the established phase boundaries. Then, the pressure is increased in small steps, to follow these identified signatures as we leave the region currently accessible to thermodynamic probes. This strategy can be confidently applied in the pressure range below $p_c$, as the magnetoresistance traces are highly self-similar. At and above $p_c$, however, they strongly change shape, and as no thermodynamic baseline exists at $p > p_c$ and $B > 30$ T, the sample enters uncharted territory.

First, we consider ambient-pressure measurements in absence of the pressure medium. Indeed, the samples fabricated onto the diamond are in excellent agreement with previously published results on chip-based crystalline devices (Fig. 1)[13]. The onset of nematic behavior at $B^* \approx 28$ T (highlighted by a magenta dotted line) is signaled by a hysteretic step-like transition with a sudden strong enhancement of the in-plane MR anisotropy. One in-plane direction exhibits a drop in the resistivity, while at the same time, the orthogonal direction shows an increase. The orientation of the in-plane anisotropy had been shown to follow the alignment of the nematic director by a small in-plane magnetic field[13]. In solenoid magnets, such an experimental situation is typically achieved by rotation of the sample with respect to the field axis. The limited space of the setup used, however, did not allow a rotation of the pressure cell during the experiment. To align the

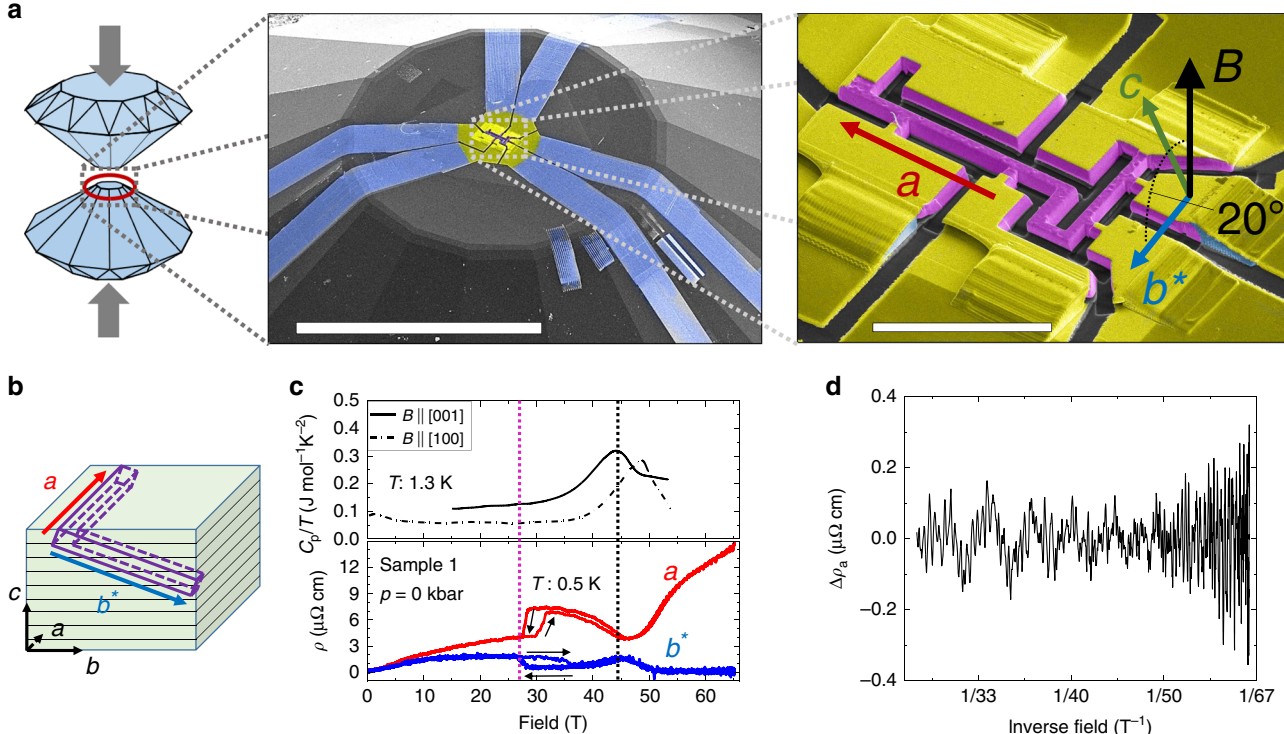

**Fig. 1 High-field magnetotransport of CeRhIn₅ measured in a DAC. a** Sketch of a typical DAC configuration, with a zoom-in electron-microscope false-color picture of the diamond culet (scale bars represent 500 and 30 μm). FIB-deposited platinum tapes (blue) connect the sample space with the outside leads. A slice of CeRhIn₅ covered with a 100 nm gold film (yellow) was microstructured by the use of a gallium dual beam FIB system. The transport device (magenta) consists of six terminals with current applied along the $a$- and $b^*$-direction (see main text), marked by red and blue arrows, respectively. **b** Orientation of the micro device with respect to the layered crystal at a deliberate tilt of $\theta = 20°$ (for further images of the devices see Supplementary Fig. 1). **c** (upper panel) Specific heat data from Jiao et al.[12]. **c** (lower panel) MR for both current directions (red and blue). Magenta- and black-dotted lines mark the nematic onset field $B^*$ and the AFM suppression field $B_c$, respectively. **d** Background subtracted resistivity $\Delta\rho_a$ plotted versus inverse field exhibiting SdH oscillations, obtained from the decreasing part of the field sweep in (**c**).

nematic order parameter in our pressure experiment, micro-devices were cut from the parent crystal at a deliberate $\theta = 20°$ misalignment angle with respect to the layered crystal lattice (see Fig. 1b). The observed in-plane resistivity anisotropy is in agreement with previous results under ambient-pressure conditions[13].

Owing to this special field configuration, however, one of the bars we call $b^*$ probes a geometric mixture of in- and out-of-plane resistivity ($\rho_{b^*} = \sqrt{\rho_b^2\cos^2(20°) + \rho_c^2\sin^2(20°)}$), while the other bar senses a pure in-plane resistivity $\rho_a$. The resistivity $\rho_c$ of CeRhIn₅ in high magnetic fields perpendicular to the Ce planes is lower than for the in-plane directions, which is consistently reflected in the lower resistivity of the deliberately tilted $b^*$ leg (see Fig. 1c)[15].

AFM order has been reported to subside at a field of about $B_c \sim 51$ T[12–14]. In this field range, we observe a peak in the $b^*$-transport channel and a dip for the $a$-direction, respectively. While the latter exhibits metallic behavior superimposed with magnetic quantum oscillations there is almost no resistivity signal to be measured along $b^*$. The clean Shubnikov-de Haas oscillations in high fields in Fig. 1c further confirm the high quality of the samples. Figure 1d shows the oscillating part of the resistivity after subtraction of the slowly varying background. The observed frequencies agree with previous de Haas-van Alphen oscillations on macroscopic crystals (see Supplementary Fig. 2) and verify an unchanged electronic state for the pressure microdevices[11]. Reproducibility of the data is particularly important given the novel and challenging character of this

experiment. To address this, we recorded a comprehensive set on three samples of similar device design and orientation in different DACs. All samples consistently support the main results (see Supplementary Information).

Figure 2 shows data obtained at low temperatures for sample 2 at eight different pressures of up to 37 kbar. We present additional data for a near duplicate device covering pressures of up to 24 kbar in Supplementary Fig. 3. The resistance noise levels did not increase under pressure, and the overall data quality remains remarkable for a good metal measured in pulsed fields in a DAC. We indicate three features evolving with pressure. (1) Gray triangles: a shoulder-like resistivity increase at $B_M$ originating from a metamagnetic transition; (2) magenta shaded areas: a sudden anisotropy enhancement related to the nematic state; (3) black squares: a shoulder-like reduction of MR for high pressures, beyond which the high-field/-pressure MR exhibits behavior we associate with magnetic order.

**Field-pressure evolution of the nematic response**. Here we examine the state with nematic character that sets in at $B^*$ (highlighted by magenta shaded areas in Fig. 2). The step-like transition at $p = 0$ gradually becomes smaller upon increasing pressure, evidencing its pressure-induced suppression. Directly tracing the pressure evolution of $B^*(p)$ is challenging as the step itself broadens and the transport signal is anisotropic regardless of nematicity due to the $\rho_c$ admixture. Instead, we focus onto another hallmark of the transition into the nematic state, its strong first-order nature. The first-order transition appears as an

extended hysteretic region, which was found to be enhanced for micron-sized devices[15]. We quantify the hysteresis via the difference between the rising and falling field sweeps, $\rho_{up} - \rho_{down}$, for both current directions $a$ and $b^*$ (see Fig. 3a). The onset field $B^*(p)$ grows upon increase of pressure, and reaches $B^*(20\,\text{kbar}) \approx 43\,\text{T}$ (see Fig. 3b and for sample 3 Supplementary Fig. 3). The hysteresis vanishes beyond 20 kbar, and at higher pressure no clear step-like signature of the nematic transition was observed. $B^*$ disappears significantly below the maximum field reached in

this experiment, $B_{max} = 60\,\text{T}$, suggesting that it did not simply leave the observation window. Further evidence for the field-induced suppression of the nematicity can be found in the amplitude of the hysteresis. While the field-scale $B^*$ increases with increasing pressure, the hysteresis amplitude is gradually suppressed into the noise floor. This is quantified in Fig. 3b showing the hysteresis amplitude as defined by black bars in Fig. 3a. Intriguingly, the nematic behavior vanishes at a pressure consistent with a line of critical points between 17 and 23 kbar reported previously from heat capacity experiments in low fields[28].

**Enhancement of the AFM state under high pressure.** Next, we turn to the observed evidence for an enhancement of magnetism upon pressure increase. The suppression of AFM order appears as a discontinuity in the slope of the MR, as commonly observed at AFM transitions in metals[29]. In the case of CeRhIn$_5$, it is most pronounced for field aligned parallel to the planes (see Supplementary Fig. 4). $B_c(p)$ continuously grows upon increasing pressure (see black diamonds in Figs. 2, 4, and 5). Here, the boundary $B_c(p, T)$ is mapped at elevated temperature and extrapolated to estimate the zero-temperature values $B_c(p)$, using a simple power law $T_N \propto (B_c - B)^\alpha$. It describes the data well with $\alpha = 0.5$, a value close to what is predicted for a spin-density-wave type QCP[30]. $B_c(p = 0)$ values obtained in this way match previous reports of the phase boundary line for fields along the $c$-direction[12] (see Fig. 6e).

We note that while the extrapolated $B_c(p)$ leaves the accessible field window when $p \geq 17$ kbar, the AFM transition remains well observable at higher temperatures. In particular, the data robustly evidence a continuous evolution of the AFM order across $p_c$ and its presence well above $p_c$. In Fig. 4, we outline the determination of $B_c$ which is apparent in the raw data as well as in the second derivative. This prominent feature in zero field has been shown to coincide with $B_c$ determined by thermodynamic measurements such as heat capacity[12] (reproduced for comparison in Fig 1c) and magnetization[14]. Upon increasing pressure, we find significant changes of the overall magnetoresistance background that complicates the determination of $B_c$. The width of the green marks in Figs. 4 and 5 indicates the tentative error bar of our determination procedure. The results are supported by sample 3 (see Supplementary Figs. 5–10).

Hence, the AFM transition continues to higher pressures beyond the zero-field QCP, $p_c \approx 23$ kbar. This is contrary to zero

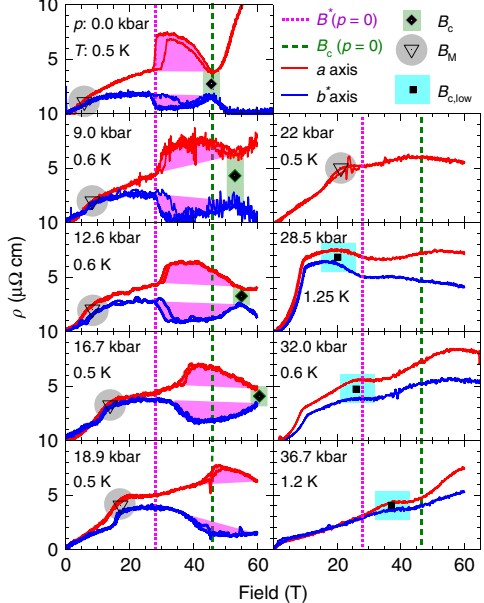

**Fig. 2 Pressure dependence of the magnetoresistivity at base temperature.** MR recorded at lowest temperatures for sample 2 at different pressures ($p = 0$ kbar was recorded for sample 1), red and blue correspond to field pulses that overlay the up- and down-sweep data for the $l||a$- and $l||b^*$-direction, respectively. The magenta dotted line marks the zero-pressure onset field of the nematic phase, $B^*(p = 0)$, and the green dashed line the zero-pressure AFM suppression field, $B_c(p = 0)$. Magenta shaded areas highlight the nematic resistivity anisotropy above $B^*$. Green diamonds mark $B_c(p)$. Gray hollow triangles mark the well-known metamagnetic transition at $B_M$ (see text). Blue squares mark a new feature we associate with field-induced magnetism, due to its temperature dependence (see text related to Fig. 6).

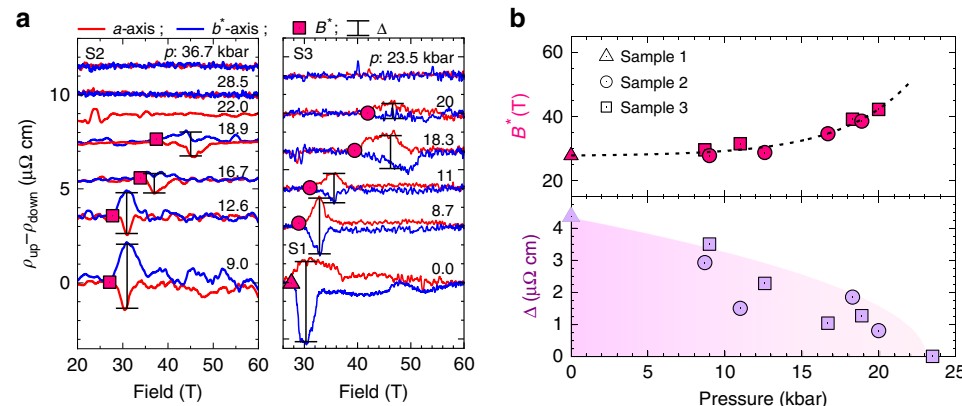

**Fig. 3 Pressure evolution of the nematic state. a** Difference between up- and down-field sweeps (data from Figs. 1c and 2 and Supplementary Fig. 3) for $l||a$ (red) and $l||b^*$ (blue), respectively. The onset field $B^*$ is marked by squares. **b** (upper panel) Critical field $B^*$ from all three samples. Dashed line is an exponential fit. Lower panel: phenomenological strength of the nematic behavior, obtained as the maximal difference (marked in (**a**) by black bars) between each pair of curves.

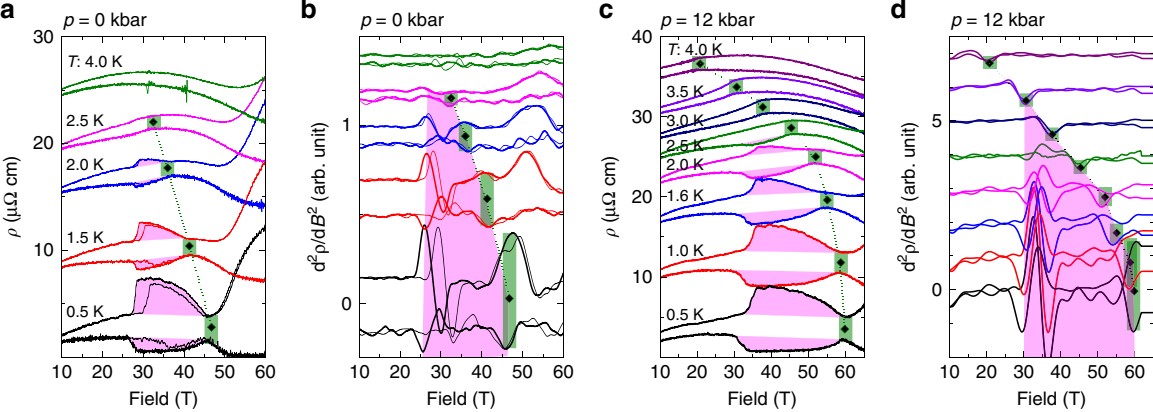

**Fig. 4 Pressure evolution of the AFM suppression field.** MR (left) and its second derivatives (right) recorded at various temperatures for sample 1 at (**a**, **b**) ambient pressure, and (**c**, **d**) $p = 12$ kbar for both current directions, $I \| a$ and $I \| b^*$; the latter has a lower resistivity at high field due to contributions of $c$-axis resistivity (see text). Magenta shading indicates nematic region. Green rectangles mark a discontinuity in the slope associated with the AFM suppression field $B_c$, which appears as a maximum/minimum in the 2nd derivative, depending on the transport direction and temperature.

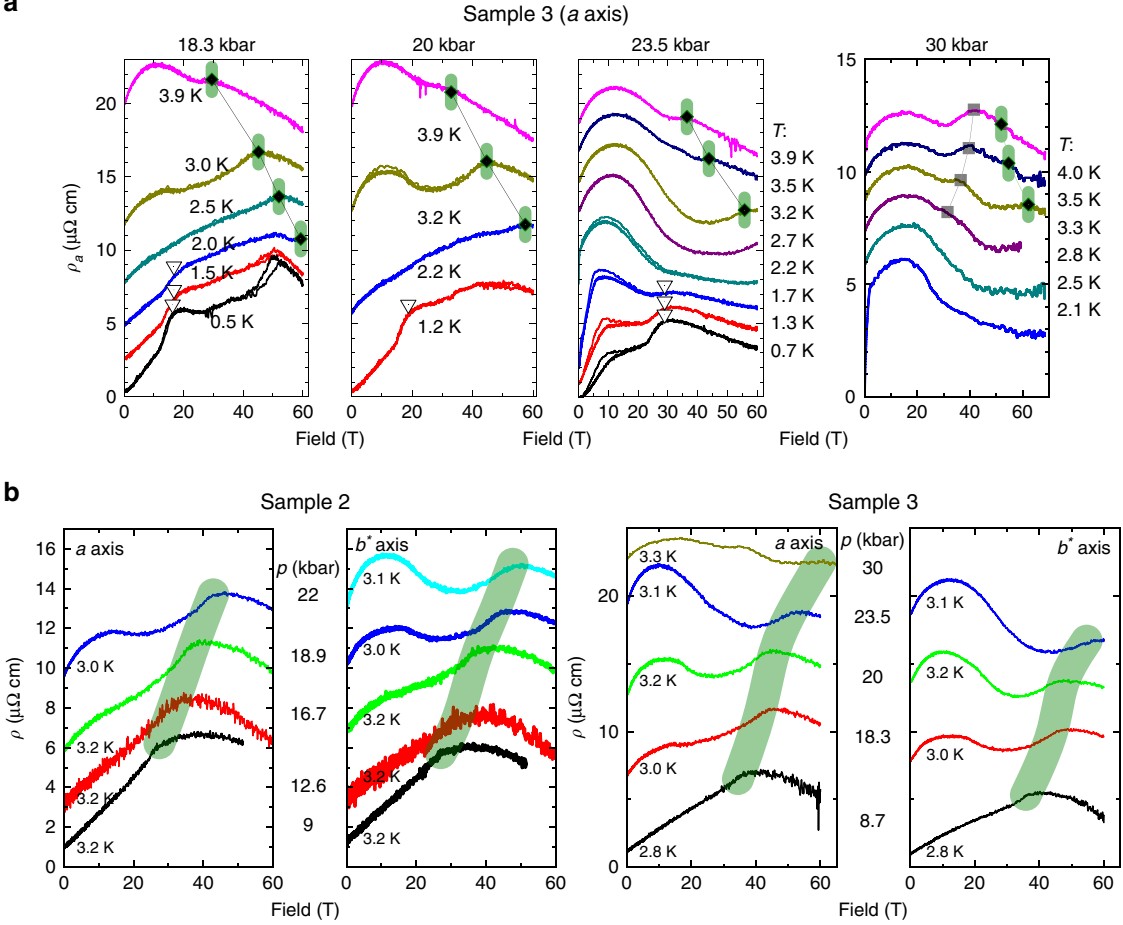

**Fig. 5 Temperature-dependent signatures of magnetic order. a** MR for sample 3 recorded at various temperatures and $p = 18.3$, 20, 23.5, and 30 kbar. The green shaded region marks the AFM suppression field $B_c$ (for a detailed comparison of samples 2 and 3 see Supplementary Fig. 5). Black squares and hollow triangles mark the onset field $B_{c,low}$ and the metamagnetic transition field $B_M$ (see main text). **b** Isothermal MR for samples 2 and 3, respectively, for different pressures at temperatures close to 3 K. This allows to directly trace the evolution of the break in slope that is known from zero-pressure measurements to correspond to $B_c(p = 0, T)$. Left and right panels exhibit the $I \| a$- and $I \| b^*$-direction, respectively. The curves are vertically offset for better visibility.

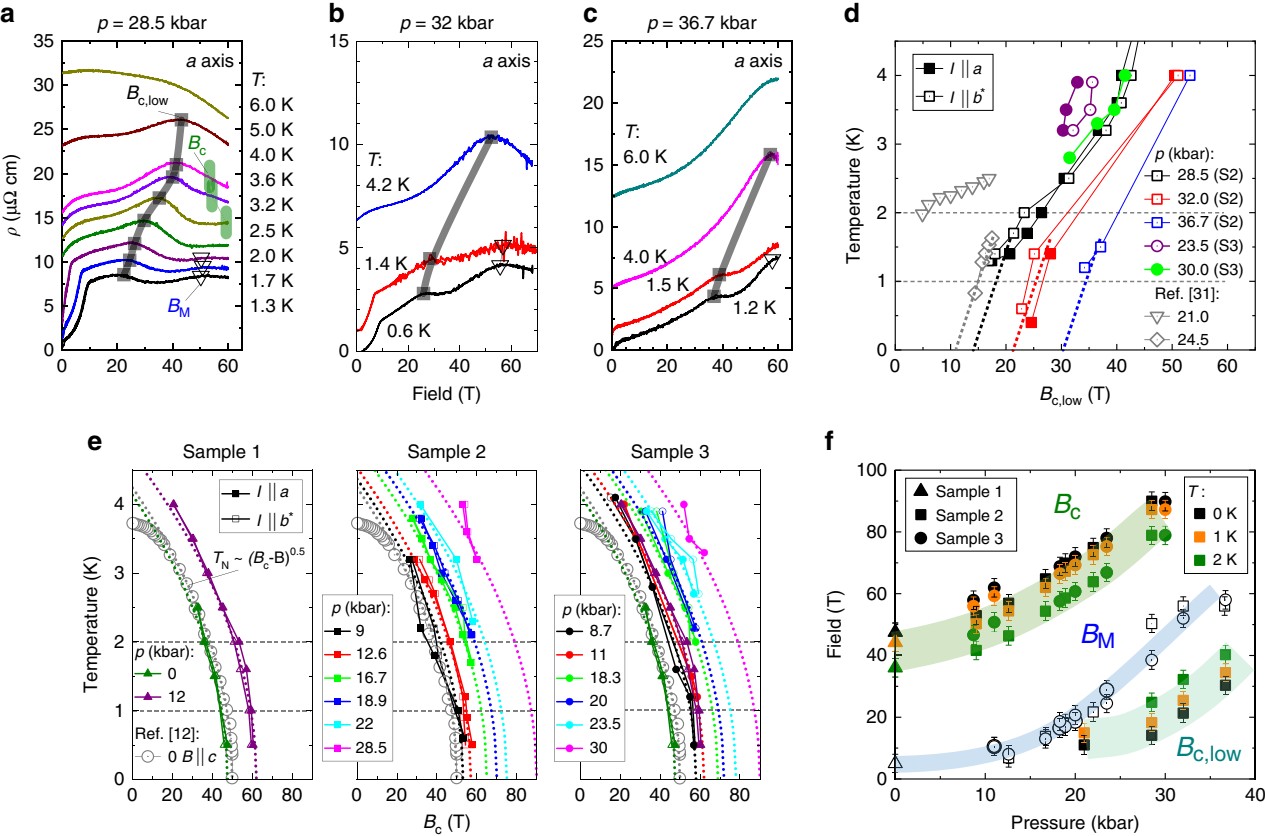

**Fig. 6 Signatures of field-induced magnetic order. a–c** MR recorded at various temperatures for sample 2 at pressures 28.5, 32, and 36.7 kbar with $I||a$-axis ($I||b^*$-data are given in Supplementary Figs. 5–10). Black squares mark the critical field $B_{c,low}$ we associate with field-induced magnetic order, shifting to higher field as temperature increases. Green vertical dashes indicate the slope change related to $B_c$. Hollow triangles mark the low temperature feature related to the metamagnetic transition field $B_M$. Note: we determined all points from 2nd derivatives, see Supplementary Figs. 6–10. **d** Pressure dependence of $B_{c,low}$. Gray data points are from previous resistivity measurements by ref. [31]. **e** Temperature dependence of $B_c(p)$. Gray circles are specific heat data for $B||c$ from Ref. [12]. Dashed lines are fits to $T_N \propto (B_c - B)^{0.5}$, see text. **f** Extrapolated critical fields $B_c$, $B_{c,low}$, and $B_M$ at 2, 1, and 0 K plotted versus pressure.

field, where magnetic order is suppressed in favor of a nonmagnetic, heavy Fermi liquid[21]. As both $p_c(B = 0)$ and $B_c(p = 0)$ bound the AFM order, they must be connected by a continuous phase boundary. One possibility, clearly ruled out by the present data, would have been a gradual suppression of $B_c(p)$, smoothly collapsing to zero at $p_c$. Instead, the presence of an AFM critical field for pressures near and above $p_c$ necessitates a field-induced reentrant magnetic order by symmetry, as the ground state in zero field at $p > p_c$ is nonmagnetic[31].

A further direct signature of the evolution of magnetism in CeRhIn$_5$ under field and pressure arises from the low-field metamagnetic transition. At this well-studied metamagnetic transition, the spin-spiral-like AFM order at zero field (AFM-I) undergoes a spin-flop transition under in-plane magnetic fields into a commensurate structure with colinear spin-configuration perpendicular to field (AFM-III)[32]. At ambient pressure, the metamagnetic transition occurs at $B_M = \frac{2\,T}{\sin(\theta)}$ owing to the strong XY-anisotropy, where $\theta$ denotes the angle between the field and the out-of-plane direction. Indeed, we observe it at 6 T as expected for $\theta = 20°$. The transition appears as an upward step for both current directions (highlighted by hollow triangles in Figs. 2, 5 and 6). As pressure increases, the resistive signature grows in magnitude, broadens and shifts towards higher fields. At pressures larger than 23.5 kbar, the associated anomaly is still discernible at the lowest temperatures (see Fig. 6a, b, f). As metamagnetism necessitates magnetic order, this observation

self-consistently provides further evidence for magnetic order above $p_c$.

**Field-induced magnetic order**. Now we turn to the structure in the magnetoresistance at intermediate fields, well below $B_c(p)$ in the high-pressure regime, $p > p_c$. A pronounced break in slope appears that persists to higher temperatures ($B_{c,low}$, marked by black squares in Figs. 5a and 6a–c). Opposite to $B_c(p, T)$, $B_{c,low}(p, T)$ increases with increasing temperature, while both increase with pressure. Qualitatively, the feature at $B_{c,low}(p, T)$ is reminiscent of the drop in resistivity at the Néel transition in zero field. It appears as a maximum in the second derivative of the MR in both current channels, and its position in field, $B_{c,low}$, moves to higher fields upon increasing pressure (see Fig. 6d). Given the similarities of the resistive signature and the thermodynamic necessity of reentrant magnetism imposed by the transition line $B_c(p)$ at high fields, we associate $B_{c,low}$ with the field-induced AFM transition. While this is a reasonable association, we caution that magnetic measurements are critical to confirm this assignment. Indeed, this reentrant scenario is well supported by previous reports in static fields at pressures beyond $p_c$[28,31].

**Discussion**
The emergence of unconventional superconductivity, magnetism, and nematicity in close proximity appears to be a unifying observation in cuprates, pnictides, and heavy-fermion systems[33–35]. In the pnictide superconductors, doping suppresses

magnetism and nematicity alike, which leads to the emergence of superconductivity around a putative nematic critical point. Our work shows this to be different in the case of CeRhIn₅ as the nematic phase moves to higher fields upon pressure increase, instead of collapsing into the zero-field magnetic QCP at $p_c \approx$ 23 kbar. Furthermore, the nematic signature weakens with increasing pressure until it vanishes very close to $p_c$ (see Fig. 3b). In light of these findings, two scenarios are possible: either the vanishing of nematicity at $p_c$ is accidental, or it points to a connection between the nematicity and the electronic reconstruction at the QCP.

First we consider the tempting scenario to correlate the vanishing of the electronic nematic behavior with the abrupt change in the Fermi surface topology that had been observed in magnetic quantum oscillation studies at lower fields[11]. Supportive of this scenario is the apparent disconnect between the magnetic state of the 4f-electrons and the nematic transition. At $B^*$ under zero pressure, no magnetic anomalies have been detected, neither in magnetization nor by torque experiments[13,14]. Although the transition occurs within the AFM region of the phase diagram, no experimental evidence points toward metamagnetism[13]. The observed huge in-plane resistivity anisotropy, $\rho_a/\rho_b \geq 6$ (see ref. [13] for the angle dependence) at $B^*$ indeed suggests a significant modification of the itinerant electron system. In addition, the subsiding strength of the anisotropy, $\Delta$, shares strong similarity with the enhancement of the effective mass as $p_c$ is approached. While under increasing pressure the effective masses are found to gradually increase until their divergence at $p_c$, the nematic order is suppressed and eventually vanishes in unison with the mass divergence. An attractive hypothesis for the commonality would be the gradual enhancement of the hybridization strength, leading to enhanced quasiparticle mass as well as weakening of electronic nematicity. This scenario, however, compares electronic reconstructions at zero field and under large fields, and if a common origin exists, it must be a field-independent reconstruction of the electronic system at $p_c$. One candidate may be topological Lifshitz-transitions in the recently reported Dirac fermions in CeMIn₅, with M = Rh, In, Co[36].

In the alternative scenario, the coincidence of $p_c$ and the pressure range at which the nematicity vanishes are purely accidental. Such a picture is supported by the pressure dependence of the critical field $B_c(p)$. While hydrostatic pressure suppresses magnetic order in zero fields, here $B_c$ increases with larger pressure until it exceeds the field window accessible to this study. At first, this growth of $B_c$ under pressure is counter to the notion that both pressure and magnetic fields suppress AFM order. However, they operate by distinct mechanisms, which counteract each other when jointly applied. Pressure suppresses magnetism in favor of delocalized 4f states by increasing the hybridization[11,28,37,38]. Magnetic field, on the other hand, favors localized moments aligned along the field in a field-polarized paramagnetic state, which clearly is the fate of any Ce-compound in the infinite field limit once the Zeeman energy surpasses any other energy scale[39–41]. While both states do not show magnetic order, they strongly differ in the degree of localization, and hence it is clear that the joint effect of field and pressure cannot be a swift suppression of AFM. This intuition may be qualitatively rationalized starting from the generalized phase diagram proposed by Doniach[42]. For Kondo lattices, pressure initially strengthens magnetism when the Ruderman–Kittel–Kasuya–Yoshida (RKKY) interactions ($T_{RKKY} \propto J^2$) dominate that favor local-moment magnetic order. At higher pressures, the on-site Kondo effect ($T_{Kondo} \propto \exp(-\frac{1}{J})$) starts to dominate. It eventually weakens the AFM order due to strong screening of the moments followed by the formation of a heavy-fermion fluid above the QCP. In a large magnetic field one would expect the Kondo screening to be suppressed, hence higher coupling strength and higher pressures are required to reach the quantum phase transition. $B_c$ should follow the pressure dependence of the RKKY scale[43], consistent with our observations.

A first step toward understanding this non-monotonic field dependence is to extend the Doniach model into the high-field region, for which no theoretical model currently exists. In Fig. 7b, we present a speculation about the main features of such a theory. With increasing $J$, it is natural to assume that the critical field of the AFM order grows with $B_c \propto J^2$. At the same time, theoretical studies of Kondo insulators suggest that a magnetic field suppresses the Kondo screening, while it enhances transverse spin fluctuations[44–47]. The associated suppression of $T_K$ with increasing magnetic field would shift the critical region, $J_c$, to higher values of $J$. Such an intuitive picture qualitatively agrees with our observations, yet it is clear that a more realistic description is required. In light of the field polarization of the conduction electrons as well as the modification of the crystal electric fields, the implicit assumption of a field-independent $J$ appears oversimplified. Further thermodynamic probes, albeit

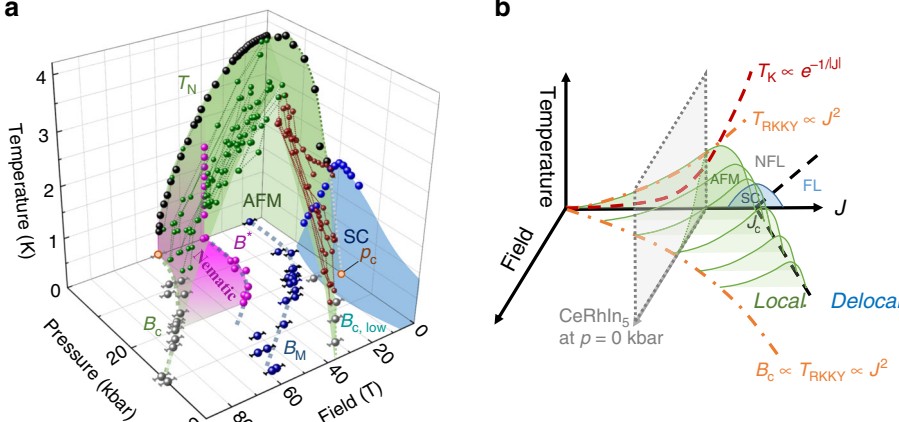

**Fig. 7 Schematic ($p$, $B$, $T$) phase diagram of CeRhIn₅. a** Experimental phase diagram of CeRhIn₅. $B^*$, $B_M$, and $B_{c,low}$ denote the nematic, metamagnetic, and magnetic transition fields and $B_c$ the AFM suppression field reported in this work. $p_c$ marks the putative QCP at 23 kbar (for the zero-temperature ($p$, $B$) phase diagram see also Supplementary Fig. 11). **b** Sketch of the ($J$, $T$) phase diagram according to Doniach et al.[42], extended by the field axis according to our experimental findings (see "Discussion").

experimentally challenging, will be required to determine the magnetic structure. In addition, resistivity measurements in the related compounds Ce(Co,Rh,Ir)In$_5$ could shed light onto the commonalities of the light bands and help to disentangle the role of the $4f$-electron state in the high-field physics.

Given the general nature of this argument, it is surprising that similar behavior is not commonly observed even in systems that are closer to an AFM QCP at ambient conditions[3]. A part of the answer may be found in the pronounced magnetic frustration of CeRhIn$_5$[48], which renders a variety of AFM orders energetically close, and thereby favors reentrant magnetism above $p_c$. Indeed, at lower fields up to 10 T detailed inelastic neutron studies have uncovered a strong field dependence of the magnetic exchange constants, evidencing the unusual role of magnetic field beyond simple Zeeman physics and the need for a field-dependent $J$ in any realistic model[22]. Part of the field dependence of the magnetic exchange will be the low lying crystal-electric-field excitations. A reported excited state of 7 meV and a very small $g$-value in the $\Gamma_7$ ground state doublet suggest a change in the orbital character in magnetic field[49,50]. Strong magnetic fields naturally change the occupied $f$-orbital, and hence modify the strength of the hybridization to the conduction electrons[18,51]. This possibility should be investigated by X-ray absorption spectroscopy or nuclear magnetic resonance studies under pressure.

In conclusion, we find the AFM suppression field $B_c$ of CeRhIn$_5$ to shift to higher fields upon pressure increase—exceeding 60 T at $p \approx 17$ kbar. Above the zero-field critical pressure $p_c$, we observe evidence for a field-induced magnetic order with an onset field $B_{c,low}$ that increases with increasing pressure. Furthermore, our magnetotransport studies show that superconductivity and nematicity reside in separate parts of the $(p, B, T)$ phase diagram. The nematic onset field $B^*$, characterized by the step-like onset of in-plane resistivity anisotropy, grows with applied pressure from 28 T at ambient conditions to around 40 T for pressures close to $p^* \approx 20$ kbar. At the same time, the anisotropy continuously decreases and vanishes completely around $p^*$. Key to understanding both the relation between quantum criticality and nematicity, as well as the anomalous phase diagram, is the fate of the Kondo breakdown in the presence of strong magnetic fields. At low fields, specific heat measurements have revealed a line of critical pressures for the suppression of AFM order between $p_{c,1} = 17$ kbar and $p_c = 23$ kbar, the critical point at which quantum oscillation studies find a delocalization transition of the $4f$ states. Does this localized-to-delocalized transition coincide with the field-induced magnetic order above $p_c$, at $B_c(p, T)$, as sketched in Fig. 7b? Or does it occur at a field-independent pressure scale of $p_c$, suggested by its coincidence with the pressure scale of the vanishing nematic state? Further theoretical and experimental efforts that contribute thermodynamic measurements will be critical to distinguish between these, or yet alternate, scenarios. Both critical end points $p_c$ and $B_c$, at which AFM order is suppressed, must be connected by a continuous line, as the transition is associated with a change in symmetry. The significant changes of the magnetoresistance beyond $p_c$ suggest a nontrivial behavior in high fields and pressure. The observed AFM phase boundary seems to deviate strongly from the commonly expected dome-like appearance, and thus hints at multiple low energy scale phenomena and potentially new correlated physics at higher pressure and magnetic field. This again emphasizes the surprising versatility of Ce-115 compounds to form a large number of almost degenerate ground states, including inhomogeneous and textured phases[22,35]. The understanding of the microscopic roots of this phenomenological observation will present a major advancement on the path of solving the strongly correlated electron problem.

## Methods

**Challenges and solutions.** DACs and custom plastic $^3$He-fridge tails and $^4$He-cryostat tails were developed at the NHMFL DC-field facility in Tallahassee, FL (USA). The high-field experiments were performed in a multi-shot 65 T magnet system at the NHMFL pulsed-field facility in Los Alamos, NM (USA). The small bore (15.5 mm) of the 65T magnet at LANL limits the overall sample size inside of the $^3$He cryostat to about 10 mm in diameter. The plastic DAC fits into this space and provides a high-pressure volume with less than 200 μm in diameter for the transport devices under pressure on top of the culet of the diamond; see the zoom-in images in Fig. 1a.

The strong forces induced by the compression of the gasket to reach high pressures above 30 kbar commonly degrade the leads fed into the sample space. This issue is naturally absent in FIB-deposited platinum leads. The FIB-deposition process is based on the ion beam induced decomposition of a Pt-containing precursor gas, methylcyclopentadienyl-trimethyl platinum. The deposited material is rich in carbon, typically around 30 at.%[52]. At the same time, the high kinetic energy of the incident ions (30 keV) amorphizes a ~20 nm thick surface layer of the diamond, breaking the C–C bonds. This allows for a chemical bonding process of the carbon-rich deposit onto the diamond. This chemical bonding results in the mechanical adhesion of FIB-deposits on diamond, compared to other approaches of metallization based on deposition and diffusion.

Measuring magnetotransport in highly conductive metallic samples such as CeRhIn$_5$ ($\rho_{xx}|_{T=0K} \approx 0.5$ μΩ cm) in pulsed fields is prone to self-heating effects due to strong eddy currents induced by rapidly changing magnetic field (LANL: pulse duration $t \approx 0.1$ s, with a rise time of 9 ms). This imposes limits on the achievable base temperature as well as on the thermal stability during the pulse. By use of FIB microstructuring, the shape of devices can be designed to minimize eddy currents. Precise control over the sample geometry on the sub-μm level enables us to tune the total device resistance into the experimentally favorable range of 1–10 Ω. This permits high-precision measurements and signal-to-noise ratios of about $10^{-3}$ with a noise of about 1 μV at a measurement frequency of 450 kHz yielding 2 nV/$\sqrt{Hz}$ noise figure (see Fig. 1c). These considerations are particularly crucial as we rely on measurements of transport anisotropy, extracted from two simultaneous measurements in the cell. We show low-field characterization data in Supplementary Figs. 12 and 13.

Combining these approaches allows us reliably to conduct multi-terminal magnetotransport measurements in a strongly constrained sample space under hydrostatic pressures of up to 40 kbar (see, e.g., Fig. 2).

**Diamond-anvil pressure cell and pressure determination.** Nonmetallic pressure cells and gaskets have been developed for pulsed-field experiments to avoid eddy current heating due to rapidly changing fields during the pulse[26]. The absence of significant heating is evidenced by the overlap of up- and down-sweep curves recorded at a temperature of 0.5 K, as shown in Fig. 2 of the main text and Fig. 1c, except for the hysteresis which is related to the intrinsic physics of CeRhIn$_5$. The use of nonmetallic cells and gaskets enables us to reach and sustain $^3$He temperatures in field of up to 65 T.

Various pressure media can be used, depending on the pressure range as well as the reactivity of that medium with the sample. For this study, we used glycerin, as it remains hydrostatic to 30 kbar at low temperature. The pressure determination is based on the detection of ruby fluorescence lines[53]. Hydrostatic conditions are monitored by measurements of the full-width-half-maximum (FWHM) of the ruby fluorescence line: FWHM < 0.3 nm for hydrostatic conditions as defined in ref. [54].

Micron-sized ruby spheres were placed inside the DAC, close to the sample so that they experience the same pressure conditions as the sample. Fluorescence was induced by a low-power 532 nm pump-diode laser. We determined the pressure in the cell, $p_{DAC}$ at room and at $^3$He temperature via optical fibers placed against the back of the diamond. In order to have a reference for the ambient pressure an additional set of spheres was attached onto a separate optical fiber and placed outside the cell at the same temperature. The difference of the fluorescence peaks, $P_1$ and $P_2$ of the ambient and pressurized ruby spheres, respectively, was used to determine the pressure via the expression: $p_{DAC} = \frac{(P_1 - P_2) \text{ nm}}{0.0365 \text{ nm/kbar}}$[55].

**Single crystals and focused ion beam (FIB) microfabrication.** We fabricated transport devices from high-quality single crystals of CeRhIn$_5$ by the application of Ga or Xe FIB microstructuring, which enable high-resolution investigations of anisotropic high-field transport. Single crystals of CeRhIn$_5$ were prepared using indium flux[8]. The samples were confirmed to have the tetragonal HoCoGa$_5$ structure by X-ray diffraction measurements and were screened by resistivity and susceptibility measurements, which showed no detectable free indium. The high quality of the samples is reflected by the very small residual resistivity and the presence of quantum oscillations in transport and thermodynamics. All devices used in this study were fabricated from 1-mm-sized single crystal of CeRhIn$_5$. The crystal was aligned by Laue diffraction, which agreed with the clearly visible tetragonal morphology of the crystal. FIB micromachining had been successfully applied to CeRhIn$_5$[13,15] and the details of the fabrication process can be found elsewhere[27]. Electrical-transport devices were fabricated directly on the culet of the diamond anvil (see Fig. 1a). Electrical contact to the sample in the cell was made via platinum leads fabricated by ion-assisted chemical vapor deposition using either Ga- or Xe-ions at currents $I_{FIB}$ between 1 and 21 nA. The CeRhIn$_5$ microstructure

was placed into the center of the cell. To this end, first a $(100 \times 20 \times 3)$ μm$^3$ slice of CeRhIn$_5$ was FIB-cut from the parent crystal and manually transferred ex-situ onto the culet without any use of adhesives or glue. Then, wedge-shaped Pt ramps were FIB-deposited on each side of the crystal slice that provides a smooth slope from the culet surface onto the crystal. Afterwards, a 100-nm thick gold layer was sputtered on top in order to improve the electric contact between the platinum and the crystal. Lastly, the excess gold was removed by FIB-milling in a negative lithography step and the sample was cut into L-shaped transport devices, highlighted by magenta color in Fig. 1a.

We conducted electrical transport measurements by a standard 4-terminal Lock-In technique with current densities of up to $1 \times 10^8$ A/m$^2$ and frequencies as high as 450 kHz. The raw data are obtained from the bare preamplified voltage response. A digital Lock-In and Butterworth filter procedure was applied afterwards in order to remove background noise. The raw MR data presented in this work were processed with the same filter parameters for consistency.

## Data availability

The ASCII data files of all important data in the figures that support the findings of this study are available in Zenodo with the identifier doi:10.5281/zenodo.3888205.

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

## Acknowledgements

We are extremely grateful for help by S.A. Crooker in the determination of the pressure. We also would like to acknowledge the technical and engineering staff, and students at the NHMFL Tallahasse and Florida State University for the design and fabrication of the plastic DACs, probes, and cryogenic equipment: R. Schwartz, M. Oliff, V. Williams, L. Riner, E. Wackes, D. Sloan, W. Brehm, D. McIntosh, A. Rubes, and R. Stanton. We also thank S. Seifert and the microstructured quantum matter group at MPI CPfS in Dresden for support in the sample fabrication process. We would also like to thank I. Sheikin for helpful discussions. The project was supported by the Max Planck Society and funded by the Deutsche Forschungsgemeinschaft (DFG, German Research Foundation)—MO 3077/1-1. P.J.W.M. acknowledges funding from the Swiss National Science Foundation through Project No. PP00P2-176789. E.D.B. and F.R. were supported by the US DOE BES DMSE program "Quantum Fluctuations in Narrow Band Systems." Work at the National High Magnetic Field Laboratory was supported by National Science Foundation Cooperative Agreements Nos. DMR-1157490 and 164477, the State of Florida, and the US DOE. A.G. acknowledges support by NNSA SSAP DE-FG-52-10NA29659. F.F.B and J.S. acknowledge support in high-field technique development from DOE BES program "Science in 100 T". We acknowledge the support of the HLD at HZDR, member of the European Magnetic Field Laboratory (EMFL).

## Author contributions

T.H., A.D.G., F.F.B., K.R.S., F.R., S.W.T., and P.J.W.M. designed research, performed research, and analyzed data. J.S., J.B.B., T.F., and M.K. provided valuable support for experiments. E.D.B and F.R. provided single crystals. All authors helped write the paper.

## Competing interests

The authors declare no competing interests.
