## [Peer Review File · Nature Communications]

Reviewers' comments:

Reviewer #1 (Remarks to the Author):

Report on manuscript NCOMMS-20-03079 Non-monotonic pressure dependence of high-field nematicity and magnetism in CeRhIn5

This paper describes the experimental study of the different phases of the heavy fermion antiferromagnetic and superconducting system CeRhIn5 tuned with a combination of high pressure, low temperature and extremely high pulsed magnetic fields

This is an impressive feat and the described technique using a FIB to prepare the sample with a suitable geometry but also to perform much of the setting up of the sample on the diamond anvil is innovative and impressive. This technique has undoubtedly huge potential for many studies. The paper reports 3 major findings. 1) The reinforcement of antiferromagnetic order with pressure as the field induced critical point shifts to higher fields as pressure is increased. 2) the emergence of a re-entrant antiferromagnetic state in applied field at pressures above the critical pressure. And 3) the disappearance of the field induced anisotropy of the in plane resistivity, previously attributed to a nematic phase, at a pressure close to P_c

I find the first 2 points convincing. The authors provide an explanation for the re-entrant magnetic phase which could be valid, though it is not clear from the discussion whether this "extended Doniach diagram" has a solid theoretical basis or is a phenomenological picture drawn from this particular experimental result. It is also not immediately clear to me why, as indeed a magnetic field will weaken the Kondo screening, it will not similarly weaken the RKKY interaction, as the same exchange J is at the basis of both effects. And indeed if it were so simple many more similar cases should be found which is not the case as the authors point out. The authors should clarify this part of the discussion.

The experimental result supporting the 3rd point is also solid. However I don't find the discussion totally convincing and I am surprised that this point is put forward as the main result of the paper, probably because nematicity is a fashionable keyword at the moment. The fact that this phase vanishes close to P_c suggests to me that, contrary to what the authors state, there could be a connection between the AFM order and the "nematic state". In the archetypal system Sr3Ru2O7 it was shown that the 'nematic behaviour' could be explained by the magnetic field control of the SDW domains. The magnetic structure of CeRhIn5 is complex, and it's true that no strong effect is seen in the magnetization, but can a similar explanation really be ruled out? In fact wouldn't it be surprising that a magnetic field with an in-plane component would have no effect on the in plane magnetic anisotropy with a natural feed back on the resistivity. I believe the authors should at the minimum open the discussion to a connection between magnetic order and the behavior so far attributed to nematicity.

My final criticism is the lack of a clear figure regrouping all the main results. This in principal is contained in the 3D plot of fig 7, where the authors have made a laudable effort to include the experimental points, but I would like in addition or instead a 2D (p - B) plot at zero or base temperature, with all the different phases clearly labeled. This would capture the essential physics and results, as the temperature effects are not really discussed except to obtain an extrapolation to low temperatures.

I think with substantial modification of the discussion the paper certainly warrants publication, and the innovative and impressive techniques used to perform these challenging experiments make the paper suitable for Nature Communications

Reviewer #2 (Remarks to the Author):

The manuscript by Helm et al. reports a detailed magnetoresistivity measurement of CeRhIn5 under high magnetic field and high pressure, which is indeed a challenging experiment. Based on these results, the authors constructed a three dimensional T-B-p phase diagram, showing that the critical field $B_c(p)$, required to suppress the AFM order, increases with increasing pressure, which is surprising because pressure suppresses the AFM order around 23kbar. Furthermore, the authors also found that the so-called nematic phase shifts to higher field with increasing pressure and then suddenly vanishes around 20 kbar. In my opinion, there is short of evidence to support such a conclusion and the current manuscript is not suitable for publication in Nature Communications.

Below are some detailed comments.

The starting and central point of this work is the field-induced nematicity. As addressed by the authors in the introduction, the broken C_4 symmetry is naturally expected to be reflected by a small lattice distortion. If this is the case, the tetragonal lattice symmetry changes to orthorhombic above B^* . Can one still be able to define the broken symmetry/in-plane anisotropy as nematicity? If not, the direction of the conclusion is questionable. In addition, even in a tetragonal system, why should the resistivity between a and b^* axis be the same?

The crucial point is that the determination of the critical fields of $B_c(p)$ is not clear and sometimes sounds arbitrary for me. For example, there is no obvious feature for B_c in Fig. 6a. In Fig. S6, the situation is even worse for $B//a$, as B_c is defined as the peak, dip or shoulder in the second derivative of $\rho(B)$. A self-consistent definition is needed to determine $B_c(p)$. If we look at the magnetoresistivity curves in the figures, typically each data set shows several "features" and therefore it is difficult to pick up one as the critical point. As a result, the derived phase diagram is questionable and it has to be confirmed by other experiments.

The experimental signatures of the last two points in the phase diagram (around 80T and 30kbar along the B_c line) are missing. From the figures, it seems that all the measurements end at 60 T. These two data points are critical. Without them, the AFM phase may abruptly vanish around p_c , like the nematic phase.

In the discussion part, scenario 1, the authors suggest that the nematic transition occurs in the light bands, without changes in the $4f_1$ states. How is it compatible with the observed changes of the dHvA/SdH frequencies from heavy quasiparticle bands at B^* ? The heavy quasiparticles should have significant contributions to the resistivity.

Some minor points:

Page 4, right column, line 12. There is a typo "showcase";

Page 6, fig. 5, what are the humps around 10 T;

Page 6, left column, line 6. The definition of "reentrant magnetic order" above p_c is misleading, since AFM phase does not enter above p_c in zero field and the AFM phase dome is continuous in Fig. 7.

Alternatively speaking, p_c should be a function of B.

Page 9, left column, line 15 to the bottom, there should be a typo in " $B_c(p, T)$ and $B_c(p, T)$ "

Reviewer #3 (Remarks to the Author):

The manuscript entitled 'Non-monotonic pressure dependence of high-field nematicity and magnetism in CeRhIn5' by Helm et al. reports on magnetoresistivity measurements carried out at extreme

conditions on the correlated metal CeRhIn₅. Notably, the measurements were carried in the challenging combination of pulsed high magnetic fields up to 60 T with high pressures of up to 40 kbars at low temperatures. CeRhIn₅ is a prototypical heavy fermion material in which the hybridization of localized f-electrons with conduction electrons results in a highly tunable strongly correlated metallic quantum state from which a multitude of quantum matter states ranging from complex and frustrated antiferromagnetism, unconventional superconductivity, and an electronic state with a large nematic susceptibility emerge. As such it is a crucial model system for our understanding of metallic quantum matter more generally. In their measurements Helm et al. address the relationship of two distinct magnetic quantum critical points (QCPs) that can be accessed by tuning the antiferromagnetic ground state of CeRhIn₅ with pressure and magnetic fields, respectively. The unconventional superconducting state in CeRhIn₅ appears around the pressure-accessible QCP, whereas the electronic state with high nematic susceptibility is observed as a function of applied magnetic field. Because in other quantum materials that exhibit superconductivity and electronic nematic order these states are typically observed in the vicinity of the same QCP, clarifying the relationship between these two quantum states in CeRhIn₅ is crucial for our understanding of quantum matter. As such, the results reported by Helm et al. in this manuscript are highly-relevant for the scientific community working on quantum materials and certainly justify publication in Nature Communications.

Further, I would like to point out that the data reported in this manuscript appears to be of high quality, which is particularly notable considering the complex sample environment (high field & high pressure at low temperature) required to obtain the data. This alone, makes the reported work highly remarkable and interesting to the community, as this technical achievement will likely open up ground-breaking studies of other relevant metallic quantum materials. To summarize, I strongly support the publication of this manuscript. However, before it can be published, I would like to ask authors to consider my comments and suggestions listed below.

Spelling and other simple corrections:

1. In the abstract, CeRhIn₅ is spelled twice without the subscript for the number 5. Please correct.
2. In the caption of Fig. 1, the expression "specific heat capacity" is used. I think that this quantity is either referred to as "heat capacity" or as "specific heat". Please correct.
3. Figure 7, panel (a). The label '0' of the pressure and temperature axis overlap. I would correct this as it makes the figure hard to read. Further there is also a purple region in the figure that does not have a label in the figure or the caption. This should be the fluctuating nematic phase, and should be labelled.
4. Section "Field-induced magnetic order". In this section Ref. [26] cites previous reports of magnetic order being stabilized by magnetic field. I think that Ref [24] is the first report of such behavior. Please correct.
5. Ref [43] is used to state that CeRhIn₅ exhibits frustrated magnetic interactions. However, this neutron diffraction study mostly points out that the ordered moment even at ambient pressure is strongly suppressed because of strong Kondo interaction. The manuscript that reports frustrated magnetic interactions are Pinaki Das et al. Phys. Rev. Lett. 113, 246403 (2014) and D. M. Fobes et al. Nature Physics 14, 456–460 (2018). Please correct.

More general comments that need to be addressed:

1. The entire text is rather lengthy and is at times tedious to read. I would strongly recommend to try to shorten some parts.
2. The term "nematic phase" is used rather loosely throughout the entire text. For example, in the abstract, it is not stated that the nature of the nematicity is electronic. Further, it should be clarified at the beginning that this phase does not spontaneously exhibit electronic nematic symmetry breaking,

but that the symmetry needs to be explicitly broken via a small magnetic field component in the tetragonal ab -plane of CeRhIn_5 . As such, the phase is not a electronic nematic phase per se, but a phase with a large electronic nematic susceptibility. This is why the experiments, for example, require a special orientation of the sample in the DAC as shown in Fig. 1. This should be clearly introduced in the introduction, and used in that sense throughout the text.

3. In the discussion as well as in Fig. 7(b) a relationship between the applied magnetic field and the RKKY interaction is used. I would like to point out that in Fobes et al. Nature Physics 14, 456–460 (2018) it is shown that at already small magnetic field (compared to the fields used in this study) the nearest-neighbor exchange terms change by about 30% compared to zero applied fields. This means that the field axis and the “ J ” axis in Fig. 7(b) can’t be perpendicular, and the true phase diagram is likely more complicated. This is also what the authors observe. In addition, the RKKY exchange is highly frustrated and competes with a substantial induced Ising anisotropy due to the applied field (see same paper). I think this needs to be considered here.

4. The electronic phase above B^* shows a large nematic susceptibility. However, it is clearly connected to the observed AFM phase. Notably, the nematic phase seems to only exist within the AFM phase. Further, similarly to the electronic behavior, the magnetic phase AFM III (or up-up-down-down phase) that coexists with the fluctuating nematic phase (due to the tilted magnetic phase) breaks the same C_4 inversion symmetry by aligning all spins perpendicular to the applied field. This is, for example, also discussed in Ref. [18], which states that likely the symmetry coming from the localized f -electron wave function is transferred onto the conduction electrons via Kondo coupling. Similar ideas are suggested in Fobes et al. Nature Physics 14, 456–460 (2018). I think this possibility should at least be discussed in the manuscript, as the strong relationship between magnetism and the fluctuating nematic phase cannot be neglected.

Dear reviewers,

We would like to thank you for your time and effort to review our manuscript “Non-monotonic pressure dependence of high-field nematicity and magnetism in CeRhIn₅”. It is gratifying to see your positive impression about the high quality data in likely the most challenging experiment we ever undertook. Certainly, it is fair to say that this is a tour-de-force effort to chart out completely untouched parts of the phase diagram of the 115-heavy fermions. Before addressing the comments and concerns one-by-one below, we would like to address two themes echoed in the reviews as well as our own discussions with colleagues in the field.

A. Interpretation of resistive signatures. We fully agree with comments about the interpretation of the resistive signatures. By themselves, a single dataset can display other resistive anomalies than the ones selected to draw the phase boundary. This is natural as resistivity, being a non-thermodynamic probe, can only trace the positions of phase transitions indicated by complementary thermodynamic techniques. In the paper, we lay out the logic of identification of these signatures: Starting from previously reported results from zero-field and zero-pressure thermodynamic measurements, such as specific heat and magnetization, we repeat these boundary measurements to identify the resistive anomalies associated with the known transitions. Then, we incrementally increase the hydrostatic pressure and, under the reasonable assumption that the anomaly shape does not change too rapidly under pressure, trace the self-similar transitions with field. In the high-pressure range, these features can be hardly discernable and may appear arbitrarily picked. However, as we show fully in the supplement, the key aspect of these phase-boundary points is not their signature per se, but the continuity of the signatures as a function of pressure and field.

The experiment we present truly enters an unexplored regime in this material class. Under the extreme conditions probed in this experiment, all energy scales associated with the Kondo effect, the RKKY interactions, the crystal-electric-field splitting and the Zeeman terms become comparable, merging the system into a highly complex, strongly coupled interacting system. One cannot treat the factors independently and as weak perturbations anymore, as demonstrated by the highly non-monotonic pressure-field dependence of the Néel boundary $B_c(p)$. Currently, to the best of our knowledge, no theoretical predictions for the magnetic or electronic structure under these conditions exist. It is our hope that this rich data set, which will be fully made available in ASCII form, sparks theoretical interest from which testable predictions are born. By no means do we expect our qualitative “extended Doniach” picture, while capturing some aspect of the relevant physics, to be the last word on this. We see it more as a motivating starting point to report that such regimes can be accessible at all, in particular to inspire and challenge other experimental efforts to follow up on this with thermodynamic probes, as reviewer 2 called for. To be realistic, however, it is fair to say that resistivity anisotropy at 30 kBar, 60 T and 0.5 K was difficult. Thermodynamic probes in this regime are a sizable challenge. To motivate such efforts we first need more theoretical insights into this limit of heavy fermions. which we hope to seed with this work.

B. Nematicity. We did not consider the question of the nematic nature of the phase above B^* discovered by other groups and us as central to this paper. In a previous publication [F. Ronning et al., Nature 548, 313-317 (2017)] we laid out the experimental evidence for its nematic character. The main result of this paper here is to trace the field scale, $B^*(p)$, into the high-pressure regime. We have clearly demonstrated that resistivity anisotropy is the best-suited tool to detect this transition, regardless of its microscopic origin. For those

readers believing in a metamagnetic origin, despite currently no experimental evidence pointing to it, this paper is equally valuable as it presents the pressure evolution of the metamagnetism. We have kept the theoretical discussion intentionally clear of the nematic phase and focus mostly onto the magnetic state. Indeed, we are completely open to any alternative interpretation of this phase and welcome any attempts of finite- q probes to identify a spin reordering, if present. However, given the technical novelty and richness of the already presented paper, we hope the reviewers understand why we do not extend the manuscript by discussing the previously published evidence for nematicity here again. The current study does not add any new information on this matter. The upward slope of $B^*(p)$ and its eventual vanishing is the main result, and it naively is compatible with both an electronic nematic based on the pressure-enhanced hybridization as well as the pressure-enhancement of a magnetic coupling that shadows the AFM order.

Again, we thank you for your time to review this paper and for the thoughtful comments addressed in the following:

Reviewer #1 (Remarks to the Author):

Report on manuscript NCOMMS-20-03079 Non-monotonic pressure dependence of high-field nematicity and magnetism in CeRhIn5

This paper describes the experimental study of the different phases of the heavy fermion antiferromagnetic and superconducting system CeRhIn5 tuned with a combination of high pressure, low temperature and extremely high pulsed magnetic fields. This is an impressive feat and the described technique using a FIB to prepare the sample with a suitable geometry but also to perform much of the setting up of the sample on the diamond anvil is innovative and impressive. This technique has undoubtedly huge potential for many studies.

The paper reports 3 major findings. 1) The reinforcement of antiferromagnetic order with pressure as the field induced critical point shifts to higher fields as pressure is increased. 2) the emergence of a re-entrant antiferromagnetic state in applied field at pressures above the critical pressure. And 3) the disappearance of the field induced anisotropy of the in plane resistivity, previously attributed to a nematic phase, at a pressure close to P_c

I find the first 2 points convincing. The authors provide an explanation for the re-entrant magnetic phase which could be valid, though it is not clear from the discussion whether this “extended Doniach diagram” has a solid theoretical basis or is a phenomenological picture drawn from this particular experimental result. It is also not immediately clear to me why, as indeed a magnetic field will weaken the Kondo screening, it will not similarly weaken the RKKY interaction, as the same exchange J is at the basis of both effects. And indeed if it were so simple many more similar cases should be found which is not the case as the authors point out. The authors should clarify this part of the discussion.

We fully agree with this assessment, and while we tried to be as clear as possible about this point before, we have emphasized it more in the revised manuscript. Indeed, at present the extended Doniach diagram is more a cartoon than a proper theory. Nevertheless, it qualitatively captures the main point of a field-induced suppression of the AFM ordering through a weakening of the Kondo interaction. The Kondo suppression acts less through a field-dependence of the coupling J , but rather through the Zeeman splitting of the doublet, which suppresses the spin-exchange. We have added a statement about this into the text.

The experimental result supporting the 3rd point is also solid. However I don't find the discussion totally convincing and I am surprised that this point is put forward as the main result of the paper, probably because nematicity is a fashionable keyword at the moment. The fact that this phase vanishes close to P_c suggests to me that, contrary to what the authors state, there could be a connection between the AFM order and the "nematic state". In the archetypal system $Sr_3Ru_2O_7$ it was shown that the 'nematic behaviour' could be explained by the magnetic field control of the SDW domains. The magnetic structure of $CeRhIn_5$ is complex, and it is true that no strong effect is seen in the magnetization, but can a similar explanation really be ruled out? In fact wouldn't it be surprising that a magnetic field with an in-plane component would have no effect on the in plane magnetic anisotropy with a natural feed back on the resistivity. I believe the authors should at the minimum open the discussion to a connection between magnetic order and the behavior so far attributed to nematicity.

One of the main results is the pressure and field evolution of this phase, not that it is a nematic. We do have a solid scientific argument in which we present strong evidence for the nematic character of this phase [F. Ronning et al., Nature 548, 313-317 (2017)]. Regardless of the microscopic origin and the question of nematicity (finite- q vs. $q = 0$), it is evident that resistivity anisotropy is the best experimental characteristic to measure the onset of this phase above B^* . Unlike the faint signatures in other probes, it is clearly detectable as a huge in-plane anisotropy as we also show in our data. Thus, if indeed a highly exotic magnetic transition is at the origin, our results reported here remain perfectly valid.

The paper at hand only reports the pressure dependence of B^* , it does not provide any additional evidence for or against the nematic character of this phase. We hope for your understanding that we would prefer not to further lengthen this paper by reiterating previous zero-pressure publications.

For the rebuttal discussion here, the transition is invisible in longitudinal magnetization as well as torque, thus placing severe limitations on the change of magnetic structure at B^* . It means that neither the net magnetization nor the orthogonal component (as measured by torque) changes within experimental resolution, while a 10-fold resistivity anisotropy spontaneously develops within the plane. We are not aware of any metamagnetic transition that is undetectable by magnetic measurements yet so dramatically impacts conduction. In fact, $Sr_3Ru_2O_7$ provides the perfect counter-example:

$\text{Sr}_3\text{Ru}_2\text{O}_7$ indeed features a huge magnetic anomaly [R. Borzi et al., PRL 92, 216403 (2004)] at the “nematic transition”, which is now known to be a spin density wave (see figure above). The complete opposite happens in CeRhIn_5 (see figure below). We have demonstrated a stringent upper limit of 1 % of a maximal change in the magnetic susceptibility given by our noise level, in contrast to the factor of 30 (!) increase in $\text{Sr}_3\text{Ru}_2\text{O}_7$, which is not atypical for a SDW formation.

As we have shown in [F. Ronning et al., Nature 548, 313-317 (2017)], the magnetic torque crosses through the B^* line without any discernable features. Similarly, the longitudinal magnetization does not show any feature in this field range [T. Takeuchi et al., JPSJ 70, 877-883 (2001)]. We took great care, inspired by the example of $\text{Sr}_3\text{Ru}_2\text{O}_7$, to investigate possible changes in the magnetic structure, and so far have not discovered any. In addition, strong microscopic evidence against a metamagnetic transition comes from In-NMR, which finds at B^* no change of the internal magnetic field at the In-1 site, but a strong response in the Knight shift [G.G. Lesseux et al., arXiv:1905.02861v2]. This strongly supports a scenario of an

electronic structure change, concordant with the quantum oscillations, without any change in the magnetic order.

We believe that if this was a metamagnetic transition, it would be much more exotic than an electronic nematic state. Something dramatic happens to the in-plane conduction as the conductor develops suddenly a 10-fold in-plane anisotropy. At the same time, the macroscopic magnetic properties remain completely unchanged, meaning that the putative spin reorientation keeps the total magnetization $M(H)$ and the susceptibility above and below the transition *exactly* identical; and the internal fields seen by NMR remain unchanged. One cannot but wonder what energetics drive such an exotic spin transition, as we can rule out the usual mechanism of minimizing Zeeman energy given that M does not change.

My final criticism is the lack of a clear figure regrouping all the main results. This in principal is contained in the 3D plot of fig 7, where the authors have made a laudable effort to include the experimental points, but I would like in addition or instead a 2D (p-B) plot at zero or base temperature, with all the different phases clearly labeled. This would capture the essential physics and results, as the temperature effects are not really discussed except to obtain an extrapolation to low temperatures.

We agree with this comment, in particular as we hope to spark more theoretical quantitative work on this. We do feel though that the 3D plot is helpful to qualitatively understand the complex landscape. If you agree, we would prefer to keep the 3D version in the paper, and publish the 2D projection in the supplement (We added a new section 5 and Figure S11 in the into supplement). Regardless, we will make all data, raw and extracted, fully available on a repository so that the data points can be directly used for quantitative analysis.

I think with substantial modification of the discussion the paper certainly warrants publication, and the innovative and impressive techniques used to perform these challenging experiments make the paper suitable for Nature Communications

Reviewer #2 (Remarks to the Author):

The manuscript by Helm et al. reports a detailed magnetoresistivity measurement of CeRhIn₅ under high magnetic field and high pressure, which is indeed a challenging experiment. Based on these results, the authors constructed a three dimensional T-B-p phase diagram, showing that the critical field $B_c(p)$, required to suppress the AFM order, increases with increasing pressure, which is surprising because pressure suppresses the AFM order around 23kbar. Furthermore, the authors also found that the so-called nematic phase shifts to higher field with increasing pressure and then suddenly vanishes around 20 kbar. In my opinion, there is short of evidence to support such a conclusion and the current manuscript is not suitable for publication in Nature Communications.

Thank you for your candid assessment. We are happy to respond to any criticism, yet it has not become clear for us which aspects exactly are missing and what “such” conclusion is? We hope to address this comment by focusing on the detailed comments below.

Below are some detailed comments.

The starting and central point of this work is the field-induced nematicity. As addressed by the authors in the introduction, the broken C₄ symmetry is naturally expected to be reflected by a small lattice distortion. If this is the case, the tetragonal lattice symmetry changes to orthorhombic above B^* . Can one still be able to define the broken symmetry/in-plane anisotropy as nematicity? If not, the direction of the conclusion is questionable.

This issue is a common misunderstanding of nematic states in general [E. Fradkin et al., Rev. Mod. Phys. 87, 457 (2015)]. In the simplest pictures, such as a Pomeranchuk instability, the itinerant electronic system finds an energetically favorable ground state by breaking a rotational symmetry. Once the rotational symmetry is broken, the total symmetry of the system is lowered and all coupling terms, forbidden in the high symmetry state, now contribute to the Free energy in the nematic state. Hence, yes, once the system is nematic, also the magnetic system and the atomic lattice will relax into a new low-symmetry structure, here an orthorhombic one. This is at the heart of the old discussion in the iron Pnictides, whether nematicity, structural transition, or magnetism is the “driving force”, as all these phenomena are inherently linked by symmetry [R.M. Fernandes et al., Nat. Phys. 10, 97-104 (2014)].

Thus, from a purely group-theoretical point of view, nematicity is completely equivalent to a structural or magnetic breaking of the symmetry. There is no discernable difference in the underlying symmetry relations. The real question is what provides the dominant change of the Free energy in the problem to drive that transition. As such, the issue is equivalent to the example of charge density waves (CDW). A CDW is a finite- q order, which breaks the translational symmetries of the crystal lattice. Hence, a CDW and a structural phase transition are related if not equivalent phenomena: phase transitions that break some translational symmetries. Yet there is an unambiguous distinction in their origin. The structural phase transition is driven either by phonon softening or by states at the Fermi level. In other words, the gain in energy originates from a modification of the low-lying bonding states in the valence band, or from gapping out the itinerant charge carriers.

The case of the nematic is completely analogous [E. Fradkin et al., Rev. Mod. Phys. 87, 457 (2015)]. Is it the magnetic system that lowers the symmetry and the electronic one follows, or vice versa. We argue that for CeRhIn₅ all magnetic measurements indicate no change at all at B^* . This suggests that any symmetry-required change in the magnetic system is minor. The transport, however, completely changes and suddenly becomes highly anisotropic (factor 10

anisotropy!). This clearly suggests that the dominant change is happening in the itinerant system, not the magnetic one.

Nevertheless, it is correct that, once the symmetry is broken, all other symmetry-allowed sample responses will occur. This includes a tiny lattice response, which has recently been observed [P.F.S. Rosa et al., PRL 112, 016402 (2019)], as well as a tiny magnetic response, so small that it has not yet been detected but has to exist by symmetry.

In addition, even in a tetragonal system, why should the resistivity between a and b* axis be the same?

It should not, and it is not as we show in Fig. 1c. The admixture of ρ_c inevitably makes them different, as we consistently detect. Nonetheless, we observe very clearly the sudden increase of anisotropy as the sample enters the nematic phase.

The crucial point is that the determination of the critical fields of $B_c(p)$ is not clear and sometimes sounds arbitrary for me. For example, there is no obvious feature for B_c in Fig. 6a. In Fig. S6, the situation is even worse for $B//a$, as B_c is defined as the peak, dip or shoulder in the second derivative of $\rho(B)$. A self-consistent definition is needed to determine $B_c(p)$. If we look at the magnetoresistivity curves in the figures, typically each data set shows several “features” and therefore it is difficult to pick up one as the critical point.

We understand this sentiment and agree that some features may appear to be arbitrarily picked, and also echoed this in the introduction. If one just looks at some curves in a finite pressure range, indeed, there are other features of similar magnitude. It is important to realize however that those features indicating the magnetic transitions must smoothly extrapolate to the field and pressure values of the known phase transitions. This is a procedure we describe in the supplement in detail.

As a result, the derived phase diagram is questionable and it has to be confirmed by other experiments.

We completely agree that further thermodynamic measurements are highly desirable to confirm and extend the present phase diagram. Given the enormous effort that this first glimpse into the high-field / high-pressure state of CeRhIn₅ required, one can only imagine how difficult specific heat or magnetization measurements of this elusive state of matter will be. Clearly, this is outside of the scope of this already extensive piece of work.

The experimental signatures of the last two points in the phase diagram (around 80T and 30kbar along the B_c line) are missing. From the figures, it seems that all the measurements end at 60 T. These two data points are critical.

In order to improve on the accessibility of our data we revised Figure 5. We added data on sample 3 that previously was given only in the supplement. In particular, the set for 30 kbar, which was measured in a 70 T magnet system at HLD in Dresden was added. It complements/confirms the trend faintly observable in the data set for sample 2 recorded for 28.5 kbar. Both sets reveal a weak trace of the feature we associate with the transition field B_c , shifting to higher fields as we increase pressure.

Most of the data traces terminate at 60 T, hence it is correct that we could not measure the base temperature value of B_c . However, as the zero temperature values are of theoretical interest, but $T = 0$ is unreachable, all zero-temperature phase transitions and quantum critical points represent extrapolations of finite-temperature values. We are experimentally limited to 0.5 K

as a base temperature, and follow the common extrapolation of high-temperature data to $B_c(T = 0 \text{ K})$. Given the increase of the AFM order with pressure, it is correct that we cannot observe $B_c(T = 0.5 \text{ K})$ at high pressures, however we clearly observe it at higher temperatures where thermal fluctuations weaken the magnetic order. As we consistently use the same extrapolation routine discussed in Fig. 6e to extrapolate the transition line to $T = 0 \text{ K}$, regardless of whether we can access $B_c(T = 0.5 \text{ K})$ directly or not, the estimate for $B_c(T = 0 \text{ K})$ should not be significantly altered.

The assessment of lacking evidence for the AFM order above p_c is factually incorrect. As we show in Figure 5, the “kink” in the magnetoresistance, which at zero pressure is clearly confirmed to correspond to the AFM transition moves up as pressure increases. In particular, at 23.5 kBar and 3.2 K, the sample leaves the AFM state at 50 T. There is strong evidence for magnetic order above p_c at elevated temperatures, consistently from multiple samples.

In the discussion part, scenario 1, the authors suggest that the nematic transition occurs in the light bands, without changes in the 4f states. How is it compatible with the observed changes of the dHvA/SdH frequencies from heavy quasiparticle bands at B^* ? The heavy quasiparticles should have significant contributions to the resistivity.

It is not clear that the heavy quasiparticles dominate electrical transport. After all, the non-magnetic LaRhIn₅ is a significantly better conductor than CeRhIn₅. A static nematic transition changes the Fermi surfaces, and hence the scattering processes. The problem is so complex that one can argue either way. For example, it is well possible that the heavy bands do not reconstruct at B^* . A significant drop in the scattering τ near the transition could be an alternate explanation for the sudden appearance of additional quantum oscillation frequencies near B_c . Its origin may well lie in the nematic change of a close-by light band.

We critically emphasize that the 4f state is very important for the nematic state, we did not find any evidence for a magnetic change in the localized part of the wavefunction. It is possible that the itinerant part of the 4f plays a main role in the nematic transition, as suggested by the low but still enhanced Sommerfeld coefficient of CeRhIn₅.

Without them, the AFM phase may abruptly vanish around p_c , like the nematic phase.

The AFM transition is a symmetry breaking phase transition and, as such, cannot terminate in a critical end point. We do know from measurements in superconducting magnets at low fields that there is an AFM state, which consistently moves also to higher fields with higher pressures, in perfect agreement with our picture in Fig. 7 (these data points are included). By symmetry, these 2 lines must connect in a continuous way. Given that low-field measurements confirmed the AFM state for pressures above p_c , it is topologically necessary that the critical-field line $B_c(p)$ crosses the value of p_c at high fields.

The same argument is true for the nematic of course. Being a symmetry breaking phase transition, it can also not terminate in an end point. Nevertheless, as no nematicity has been observed at low fields, the phase line $B^*(p)$ is not bound to come down to zero. Instead, it is free to turn upwards, in a quasi-vertical line at p_c , and to terminate at another phase line that breaks the same symmetry. This is different for the case of AFM order, which must come down to $B = 0 \text{ T}$ eventually as it has been seen there before.

Page 6, fig. 5, what are the humps around 10 T;

This feature indeed is prominent, yet we do not have an explanation for it yet. Qualitatively, one may note that this peak grows as p_c is approached and shrinks above it, hence it may be associated with quantum critical fluctuations. Alternatively, in this low pressure range, coexisting yet phase incoherent superconductivity has been reported and it may be the fluctuation contributions which are suppressed as the field orbitally limits them.

Of course, we fully agree that resistivity in such a complex system always is full of “anomalies” and one can draft a zoo of “features”. Here, we rely on tracing those that occur at known phase transitions in zero field or zero pressure. Critically, we do not even in these cases quantitatively understand why the transitions appear in the resistivity in the particular shape they do. Qualitatively, one can argue in any direction as a loss of scattering channels reduces the resistivity and a loss of carriers increases it. This is why we focus on tracing self-similar features to construct the phase diagram. This hump has no correspondence in other phase transitions, which is why we do not include it in the phase diagram.

Page 6, left column, line 6. The definition of “reentrant magnetic order” above p_c is misleading, since AFM phase does not enter above p_c in zero field and the AFM phase dome is continuous in Fig. 7. Alternatively speaking, p_c should be a function of B .

We agree this sentence was not clear and rephrased it. Following the convention in the field, we denote the zero field position of the quantum critical point by p_c , and as such defined, has no field dependence. We discuss it in terms of the pressure dependence of $B_c(p)$, which of course one can invert to find $p_c(B)$ with $p_c(B = 0) = p_c$.

Reviewer #3 (Remarks to the Author):

The manuscript entitled ‘Non-monotonic pressure dependence of high-field nematicity and magnetism in CeRhIn₅’ by Helm et al. reports on magnetoresistivity measurements carried out at extreme conditions on the correlated metal CeRhIn₅. Notably, the measurements were carried in the challenging combination of pulsed high magnetic fields up to 60 T with high pressures of up to 40 kbars at low temperatures. CeRhIn₅ is a prototypical heavy fermion material in which the hybridization of localized f-electrons with conduction electrons results in a highly tunable strongly correlated metallic quantum state from which a multitude of quantum matter states ranging from complex and frustrated antiferromagnetism, unconventional superconductivity, and an electronic state with a large nematic susceptibility emerge. As such it is a crucial model system for our understanding of metallic quantum matter more generally. In their measurements Helm et al. address the relationship of two distinct magnetic quantum critical points (QCPs) that can be accessed by tuning the antiferromagnetic ground state of CeRhIn₅ with pressure and magnetic fields, respectively. The unconventional superconducting state in CeRhIn₅ appears around the pressure-accessible QCP, whereas the electronic state with high nematic susceptibility is observed as a function of applied magnetic field. Because in other quantum materials that exhibit superconductivity and electronic nematic order these states are typically observed in the vicinity of the same QCP, clarifying the relationship between these two quantum states in CeRhIn₅ is crucial for our understanding of quantum matter. As such, the results reported by Helm et al. in this manuscript are highly-relevant for scientific community working on quantum materials and certainly justify publication in Nature Communications.

Further, I would like to point out that the data reported in this manuscript appears to be of high quality, which is particularly notable considering the complex sample environment (high field & high pressure at low temperature) required to obtain the data. This alone, makes the reported work highly remarkable and interesting to the community, as this technical achievement will likely open up ground-breaking studies of other relevant metallic quantum materials. To summarize, I strongly support the publication of this manuscript. However, before it can be published, I would like to authors to consider my comments and suggestions listed below.

We highly appreciate your supportive comments, and in particular thank you for your notion on the data quality. It is indeed a challenging experiment, and despite the difficulties, we have reproduced the results in three different samples in different pressure cells to make sure the reported effects are intrinsic.

We also thank you for pointing out the minor spelling mistakes!

More general comments that need to be addressed:

1. The entire text is rather lengthy and is at times tedious to read. I would strongly recommend to try to shorten some parts.

We resonate with this impression, and tried to further shorten the manuscript. As you can see, it remains quite extensive. We report on the behavior of a correlated material in an extreme sample environment, using a new approach. In our view, the technical achievements are on par with the scientific findings. Initially, we considered to write separate papers about the technique and CeRhIn₅. However, these aspects are too intimately entangled. We need the low-field / high-pressure and high-field / zero-pressure physics of CeRhIn₅ to demonstrate that our combined approach actually extrapolates well into these known regimes. On the other hand,

given that resistivities are very difficult to interpret, it is critical for the reader to understand how exactly they were measured when reading our report on the high field science.

This is one reason why we chose Nature Communications as an online format with more generous length limitations, and structured the paper into a first technical and second scientific part. It is written in a way that either can be skipped if the reader is only interested in one aspect.

2. The term “nematic phase” is used rather loosely throughout the entire text. For example, in the abstract, it is not stated that the nature of the nematicity is electronic. Further, it should be clarified at the beginning that this phase does not spontaneously exhibit electronic nematic symmetry breaking, but that the symmetry needs to be explicitly broken via a small magnetic field component in the tetragonal ab-plane of CeRhIn₅. As such, the phase is not a electronic nematic phase per se, but a phase with a large electronic nematic susceptibility. This is why the experiments, for example, require a special orientation of the sample in the DAC as shown in Fig. 1. This should be clearly introduced in the introduction, and used in that sense throughout the text.

It is correct that “nematic” is used quite loosely, in the sense that for the paper at hand we treat it simply as a name for the state above B^* . The microscopic origin of nematic response in CeRhIn₅, however, is completely unclear and further, extremely challenging, experiments are required to detect its origin. It may well be a nematicity in the spin channel, in which case it would be magnetic but fluctuating. One hypothesis is that the orthogonal spin fluctuations suddenly pick a strongly preferred direction. This may strongly change directional charge transport, but would not correspond to a metamagnetic transition as there is no q -change in the problem and the static components of the magnetization are unchanged. This question of spin vs. charge is further complicated by the $4f$ moments, as they are not perfectly localized. The enhanced Sommerfeld coefficient of $\gamma \sim 70 \text{ mJ mol}^{-1}\text{K}^{-1}$ [R.A. Fisher et al., PRB 65, 224509 (2002)] is too low to fall into the true heavy-fermion category. It clearly is too large to suggest completely localized $4f$ states. One possibility could be that there is an electronic nematic ordering in that fraction of the $4f$ wavefunction that is itinerant, while the localized part is unchanged. While we do believe one of these exciting directions could be the basis of the physics, we currently have no evidence to support this. This is why we kept the paper databased without biasing the reader into a particular model of nematicity. Apparently, a proper high-field theory is required, and it is our hope to stimulate this experimentally.

The question of static nematic vs. strong nematic fluctuations is more subtle. The reason is pinning. Only the perfect infinite crystal has a strictly defined notion of spontaneous symmetry breaking. In the finite system, in the presence of surfaces and intrinsic imperfections, the rotational symmetry is always broken. This is typical for any symmetry-breaking phase transition, for example structural transitions. Quite commonly, a single crystal decays into many domains at a transition, such that the macroscopic measurable quantities retain the microscopically broken symmetry. For example, at tetragonal-to-orthorhombic transitions one commonly sees C_4 symmetry in the orthorhombic state due to domain averaging. In this case, a strain bias is usually used to ensure the system goes into one domain state, i.e. detwinning. It depends on the stiffness of the order parameter if either the bias field below the transition temperature can be removed in order to reach a truly detwinned system, or if it has to remain active to keep the system in a detwinned state.

This is both experimentally observed and theoretically expected for nematics. Consider the most prominent example for an electronic nematic, the $\nu=9/2$ fractional quantum hall state [e.g. J.P. Eisenstein Solid State Communications 117, 123-131 (2001)]. It is interesting to

note that Eisenstein never observed spontaneous symmetry breaking. The direction the state picks is locked to the sample, and has been ascribed to substrate-induced strain anisotropies. In addition, here, an in-plane magnetic field of up to 70° is required to swap the low and high conductive directions, i.e. align the nematic, which snaps back once the in-plane field is lowered. As we demonstrate XY-nematic behavior [F. Ronning et al., Nature 548, 313-317 (2017)], one expects a much softer nematic state in CeRhIn₅. Thus, it is perfectly plausible that a microscopic nematic transition occurs that is macroscopically undetectable for $H \parallel c$ as an equal amount of nanodomains compensate each other. It is an interesting property of this transition that the gain of conductivity in one direction perfectly cancels the loss in the other. Finite in-plane fields then just break the balance between domains.

However, this is a fully open question. Furthermore, a fluctuating nematic order is also compatible with the results, as has been argued by magnetostriction experiments. [P.F.S. Rosa et al., PRL 112, 016402 (2019)].

Thus, we believe this is an exciting research direction and identifying the microscopic origin of this phase will be key to a deeper understanding of this topic. However we do not see how our present experiments could add anything to this discussion, as we do not have a microscopic model for either nematic pressure dependence.

3. In the discussion as well as in Fig. 7(b) a relationship between the applied magnetic field and the RKKY interaction is used. I would like to point out that in Fobes et al. Nature Physics 14, 456–460 (2018) it is shown that at already small magnetic field (compared to the fields used in this study) the nearest-neighbor exchange terms change by about 30% compared to zero applied fields. This means that the field axis and the “J” axis in Fig. 7(b) can’t be perpendicular, and the true phase diagram is likely more complicated. This is also what the authors observe. In addition, the RKKY exchange is highly frustrated and competes with a substantial induced Ising anisotropy due to the applied field (see same paper). I think this needs to be considered here.

Indeed, the field-tuned easy-axis anisotropy will be a key component in understanding the phase diagram. In addition, crystal-electric-field effects and spin-polarization of the conduction sea will play a main role.

In a standard Kondo-lattice model following Doniach,

$$H = \sum_k \epsilon_k c_k^\dagger c_k + J \sum_i s_i S_i + B \sum_i g_1 s_i + g_2 S_i .$$

In this language, the exchange coupling term between local moments and spins and the magnetic field enter independently. Apparently, in a real material J cannot be tuned directly but only indirectly, though $J(B,p)$. Conceptually for a plot in spirit of the Doniach diagram, they are independent theoretical parameters. CeRhIn₅, upon increasing the magnetic field, may not evolve along a straight line in the plot then. Therefore, we agree that the B,p evolution of the exchange interactions is complex as suggested by Fobes et al., and our results. We have emphasized this in the reworked manuscript .

4. The electronic phase above B^* shows a large nematic susceptibility. However, it is clearly connected to the observed AFM phase. Notably, the nematic phase seems to only exist within the AFM phase.

This does not appear to be settled yet. The majority of the data on the nematic phase arises from static fields in the 45 T hybrid magnet, which does not allow to cross $B_c \sim 50$ T. For example, upon inspection of Fig. 2, $p = 0$, a very large a - b^* anisotropy is observed. Now this is a very different magnetic and electronic state above B_c , and the crystalline anisotropy has to be taken into account if one is to interpret the b^* direction. Nonetheless, the data would clearly be compatible at least with a persistence of the nematic state above B_c . We are currently also investigating CeIrIn₅, which appears to have a similar B^* transition at even higher fields, but in absence of static magnetic order. This would be perfectly compatible with a nematic transition in the spin fluctuation channel as discussed by Fobes et al. [Nature Physics 14, 456–460 (2018)].

Further, similarly to the electronic behavior, the magnetic phase AFM III (or up-up-down-down phase) that coexists with the fluctuating nematic phase (due to the tilted magnetic phase) breaks the same C_4 inversion symmetry by aligning all spins perpendicular to the applied field. This is, for example, also discussed in Ref. [18], which states that likely the symmetry coming from the localized f -electron wave function is transferred onto the conduction electrons via Kondo coupling. Similar ideas are suggested in Fobes et al. Nature Physics 14, 456–460 (2018). I think this possibility should at least be discussed in the manuscript, as the strong relationship between magnetism and the fluctuating nematic phase cannot be neglected.

Absolutely. The AFM III breaks magnetically the C_4 symmetry, and accordingly this denotes the onset of the small difference between the resistivities of the now inequivalent a , b directions [F. Ronning et al., Nature 548, 313-317 (2017)]. This is also clearly seen in the data here, however due to the b^* complication there is already some difference at even lower fields.

In fact, the AFM I - to - AFM III transition is a strong argument against a static spin reorientation at B^* . Apparently, the magnetic structure changes at $B_c^{\text{III}} \sim 2$ T completely, and accordingly a jump in magnetization is seen. This also appears as a small jump in resistance as well as the onset of some small a , b anisotropy. This suggests that the magnetic structure is, while of course coupled to the conduction system, not the dominant factor at determining the transport coefficients. The complete opposite happens at B^* , where no magnetic anomaly is observed but a 10-fold in-plane anisotropy emerges.

All of this is clearly compatible with the ideas of electronic texture, discussed by Fobes et al. We have added a discussion along these lines accordingly.

We would like to thank all referees for their time and effort to carefully review this manuscript.

Thank you,

Toni Helm, Philip Moll (on behalf of all authors)

REVIEWERS' COMMENTS:

Reviewer #1 (Remarks to the Author):

Report on revised version of manuscript 241006 by T. Helm et al.

The authors have made relatively few modifications to the manuscript but have incorporated several changes that significantly respond to some of the criticisms of the 3 referees.

Concerning whether the anomaly at B^* it is not a question of "belief" (theirs, mine or anyone else's) that this phase is Nematic. What appears dangerous to me is that after initial works that propose this solution, which I agree is plausible and perhaps even probable, it should then become an accepted truth.

However the addition of the sentence "For these reasons we shall refer to the sudden and strong field induced transport anisotropy at B^* as a consequence of nematicity" satisfies me. I would like in addition a phrase to the effect of what they write clearly in the rebuttal letter "The main result of this paper here is to trace the field scale, $B^*(p)$, into the high-pressure regime regardless of its microscopic origin"

I find on the other hand the discussion of the extended Doniach diagram still very muddled. If it is only intended as a cartoon perhaps this should be stated more explicitly, because for the moment it looks like a model. An extended Doniach diagram should have 2 possible mechanisms: a field dependence of J , or different field dependences of T_K and T_{RKKY} for a given J , and of course a possible combination of both. I had initially assumed that the model was based on the field effect on J . According to their rebuttal this is not the case, however it is stated in the text "To extend the zero-field Doniach model, the field dependence of the coupling J has to be taken into account (Fig. 7 b), ". However fig 7b implicitly assumes a field independent J (i.e. the magnetic bubble moves to the right with field for constant J). This diagram is far from trivial and raises questions. Actually the initial increase of B_c where T_N increases is trivial. To capture the physics here it is necessary to describe the increase of B_c when T_N decreases, so it is only in the small right hand part of the diagram. But then shouldn't B_c be a function of T_K as well as T_{RKKY} (just as T_N is) ? Or is it implicitly assumed that the Kondo effect is washed out with field but $RKKY$ is unchanged ? But then how do you explain the continuity of J_c ?

The authors should state explicitly what mechanism is in play in their diagram and whether or not this can explain all the results. Actually this part seems to me to show that a simple extended Doniach picture will have great difficulty capturing the physics in play here. The last part of the discussion is much clearer and basically the conclusion is that the explanation must be sought elsewhere. i.e. frustration or non trivial field dependence of coupling constants.

I think the authors have captured all the ingredients that can possibly explain the re-entrance of magnetism but it could be presented much more simply and clearly. I leave the editors to decide whether the present format is acceptable or whether it should be further improved.

The 2D figure in the supplement (fig S11) is very clear and useful. It would be nice to make a reference to it in the caption of fig7

I maintain my initial judgment that the high quality of the experimental work and interesting physics warrant publication in Nat Comm. While I find that the paper could still be improved, this can now be considered optional, depending on editorial decision.

Reviewer #2 (Remarks to the Author):

In the revised version, the authors weakened their major claims. This is fair as we are all aware of the versatility and the complexity of this material. For example, very recent NMR measurements further confirmed the rotational symmetry breaking of the electronic structure in the AFM3 phase already [arxiv:200501421], which should modify the phase diagram of CeRhIn5 and challenge the concluded role of magnetism in this work. In addition, given the criterion for the phase boundaries are not robust in experimental data, I leave the final judgement up to the editor.

Reviewer #3 (Remarks to the Author):

As I have previously stated, I am strongly support the publication of the manuscript. With the revised version and also with their detailed replies to my comments and suggestions, I recommend that the manuscript will be published as is. I have also studied the replies and changes in response to the remarks of the other referees, and would like to point out that the authors have considered them carefully as well. Overall my impression is that the revised version of the manuscript is substantially stronger than the already impressive previous manuscript. In conclusion, the impressive methodology together with the high quality of the data on a quantum material of current interest clearly deserve to be published in Nature Communications. I am convinced that the manuscript will be met with great interest by the readers of this journal.

Dear Reviewers,

We thank you for taking the time to review our revised manuscript “Non-monotonic pressure dependence of high-field nematicity and magnetism in CeRhIn₅”. We highly appreciate your time and effort to improve this paper, as well as your encouraging comments on the quality of our data. Given the novelty of the experimental approach, we particularly emphasized the reproducibility of the dataset. It is most gratifying to us that the main results of this paper were reproduced in 3 different samples and setups, despite the challenging nature of the experiment.

In the revised manuscript, we have further highlighted the discussion on the nematic state and emphasized the current discussions in the field. Currently no microscopic picture of the high-field state in CeRhIn₅ exists, and we hope that this work will spark further theoretical and experimental studies towards this goal. The debates, here as well as in the community, clearly show the need to resolve this problem. For example, new In-NMR data provides strong microscopic evidence for the absence of a metamagnetic transition at B^* while a sudden change in the Knight shift was observed [PRB 101. 165111 (2020)]. In this work, a picture based on delocalization transition due to enhanced Kondo hybridization is argued, which immediately undergoes a finite- Q charge density wave. This appealing picture, though, needs to be reconciled with the XY-character of the resistive anomaly as well as the apparent lack of magnetic in-plane anisotropy.

As more techniques extend into the high-field range, more and more pieces of the puzzle will be uncovered. Our high-field/high-pressure experimental work is one of them.

Let us thank you again for your time to review the manuscript. In the following, we will respond to the individual points one-by-one:

Reviewer #1 (Remarks to the Authors):

Report on revised version of manuscript 241006 by T. Helm et al.

The authors have made relatively few modifications to the manuscript but have incorporated several changes that significantly respond to some of the criticisms of the 3 referees.

Concerning whether the anomaly at B^* it is not a question of “belief” (theirs, mine or anyone else’s) that this phase is Nematic. What appears dangerous to me is that after initial works that propose this solution, which I agree is plausible and perhaps even probable, it should then become an accepted truth.

However the addition of the sentence “For these reasons we shall refer to the sudden and strong field induced transport anisotropy at B^* as a consequence of nematicity” satisfies me. I would like in addition a phrase to the effect of what they write clearly in the rebuttal letter “The main result of this paper here is to trace the field scale, $B^*(p)$, into the high-pressure regime regardless of its microscopic origin.”

We fully share the assessment that the microscopic picture is not clarified – and nematicity at best is one of many characteristics of the state above B^* . In fact, it is part of the fascination that this material, superficially a local moment anti-ferromagnet, exhibits so nuanced behavior. We

use the term “nematic” as a characteristic of the itinerant electronic system suddenly lowering its symmetry at the B^* transition, while the magnetic system appears unchanged.

We highly appreciate the supportive statement on the merit of our work. These observations will be valuable to create, test and refine any model of the high-field state, nematic or not. We have added the proposed sentence, and an even stronger statement, reflecting the ongoing discussion of this work in the field – a debate we strongly support and engage in. At the end of the day, the community aims to understand this complex system in high fields – and we believe that transport studies cannot definitely settle a discussion on microscopies. The results are highly compatible with an electronic nematic, but other measurements will have to confirm, or reject, this hypothesis.

We revised part of the introduction in order to be as clear as possible on the objectives of the present work (see List of changes below).

I find on the other hand the discussion of the extended Doniach diagram still very muddled. If it is only intended as a cartoon perhaps this should be stated more explicitly, because for the moment it looks like a model. An extended Doniach diagram should have 2 possible mechanisms: a field dependence of J , or different field dependences of TK and $TRKKY$ for a given J , and of course a possible combination of both. I had initially assumed that the model was based on the field effect on J . According to their rebuttal this is not the case, however it is stated in the text “To extend the zero-field Doniach model, the field dependence of the coupling J has to be taken into account (Fig. 7 b), ” . However fig 7b implicitly assumes a field independent J (i.e. the magnetic bubble moves to the right with field for constant J). This diagram is far from trivial and raises questions. Actually the initial increase of B_c where T_N increases is trivial. To capture the physics here it is necessary to describe the increase of B_c when T_N decreases, so it is only in the small right hand part of the diagram. But then shouldn't B_c be a function of TK as well as $TRKKY$ (just as T_N is) ? Or is it implicitly assumed that the Kondo effect is washed out with field but $RKKY$ is unchanged ? But then how do you explain the continuity of J_c ?

The authors should state explicitly what mechanism is in play in their diagram and whether or not this can explain all the results. Actually this part seems to me to show that a simple extended Doniach picture will have great difficulty capturing the physics in play here. The last part of the discussion is much clearer and basically the conclusion is that the explanation must be sought elsewhere. i.e. frustration or non trivial field dependence of coupling constants. I think the authors have captured all the ingredients that can possibly explain the re-entrance of magnetism but it could be presented much more simply and clearly. I leave the editors to decide whether the present format is acceptable or whether it should be further improved.

First of all, we highly resonate with these comments. In fact, the statement “*To extend the zero-field Doniach model, the field dependence of the coupling J has to be taken into account (Fig. 7 b)*” was a – poorly worded – attempt to already hint to the weaknesses in this cartoon.

The main point is that this 3D cartoon takes J , by definition, as a controllable tuning parameter that can be set independently from field or temperature, in which case it follows naturally. The problem is of course how to obtain J . Here the critical, and oversimplified, assumption is that it is simply set by pressure, $J(p)$, such that one can regard the J axis as an essential p -axis and compare to experiment. This is most likely a gross oversimplification, and the field-dependence of J must be properly considered in a microscopic theory. The main goal of Fig.7 is to display and discuss the main ingredients such a model should have.

In order to further improve our manuscript we revised the respective part of the discussion as given below in the List of changes.

The 2D figure in the supplement (fig S11) is very clear and useful. It would be nice to make a reference to it in the caption of fig7

We added the following sentence into caption of Fig. 7:” (For the zero-temperature (p,B) phase diagram see also Supplementary Fig.~11).

I maintain my initial judgment that the high quality of the experimental work and interesting physics warrant publication in Nat Comm. While I find that the paper could still be improved, this can now be considered optional, depending on editorial decision.

Reviewer #2 (Remarks to the Author):

In the revised version, the authors weakened their major claims. This is fair as we are all aware of the versatility and the complexity of this material. For example, very recent NMR measurements further confirmed the rotational symmetry breaking of the electronic structure in the AFM3 phase already [arxiv:200501421], which should modify the phase diagram of CeRhIn5 and challenge the concluded role of magnetism in this work. In addition, given the criterion for the phase boundaries are not robust in experimental data, I leave the final judgement up to the editor.

Thank you for appreciating the complexity of this material, and that a complete resolution of the high field/high pressure state is too much to ask from a challenging transport study. Clearly, we can only focus on the features we observe (and even those can be weak at times and rightfully challenged, as you mentioned). It is a fact that the transition from AFM-II to AFM-I is undetectable in transport. One may argue that there is a weak change in the powerlaw, but given the thin sliver of AFM-II in temperature, it would not lead to a trustworthy signature. This makes sense as this type of commensurability transition of the q -vector only implies a small, low- q change of the magnetic texture. The fate of AFM-II in high fields is interesting by itself, though, as we could not exclude that AFM-II takes a significant fraction of the high field phase space.

The transition of AFM-I to AFM-III, which we denote as metamagnetic transition, is detectable both in transport and magnetization, hence we can here trace it. One might argue that this transition is a rather radical moment reorientation, and, despite this, its signature in transport is rather weak. This again points to a non-magnetic origin of the clearly dramatic B^* transition in transport at which no macroscopic magnetic measurements detect a change.

This is perfectly compatible with these NMR results [arxiv:200501421]. As we stated in our nematicity paper [13], there is a small transport anisotropy starting at the metamagnetic transition, which is also compatible with our new data, see Fig.1. We fail to see though which aspect of this NMR result should *modify the phase diagram and the concluded role of magnetism in this work?*

In response to the helpful suggestion we revised the introduction, and now mention the suggested reference therein (see list of changes below).

Reviewer #3 (Remarks to the Author):

As I have previously stated, I am strongly support the publication of the manuscript. With the revised version and also with their detailed replies to my comments and suggestions, I recommend that the manuscript will be published as is. I have also studied the replies and changes in response to the remarks of the other referees, and would like to point out that the authors have considered them carefully as well. Overall my impression is that the revised version of the manuscript is substantially stronger than the already impressive previous manuscript. In conclusion, the impressive methodology together with the high quality of the data on a quantum material of current interest clearly deserve to be published in Nature Communications. I am convinced that the manuscript will be met with great interest by the readers of this journal.

We thank you for your time and effort to assess our work, and in particular for your appreciation of its technical difficulty. We sincerely hope that this data will invigorate further theoretical studies on the high-field phase of CeRhIn₅. As we hope to convey in the paper, this is not the concluding remarks on the problem but a first, phenomenological map of this state which hopefully can be refined and complemented in the future.

List of changes:

Taking into account the concerns raised by reviewer#1 and #2 and the new, very interesting NMR results by Kanda et al., suggested by reviewer #2, we further revised the introduction as follows:

- **Modification Introduction (end of second paragraph):**

“... For these reasons we shall refer to the sudden and strong field induced transport anisotropy at B^ as nematic for simplicity. Yet its microscopic origin remains a highly active area of research, and the nematic picture is constantly expanded, refined as well as challenged. The main open questions concern the explicitly symmetry-breaking role of the in-plane magnetic field [Kanda et al.], for example through a modification of the crystal electric field schemes, as well as potential changes of the microscopic magnetic ordering that might remain undetected by measurements of the averaged magnetization and torque. Here, the recent breakthroughs in pulsed magnetic field neutron scattering [Duc et al.] and other microscopic techniques would be most insightful. One of the main results of the present work is to trace the field scale, $B^*(p)$, into the high-pressure regime regardless of its origin.”*

In response to the very helpful criticism of reviewer#1 we revised the 4th paragraph of our Discussion as follows:

- **Modification discussion (4th paragraph):**

“A first step towards understanding this non-monotonic field dependence is to extend the Doniach model into the high field region, for which no theoretical model currently exists. In Fig. 7b, we present a speculation about the main features of such a theory. With increasing J , it is natural to assume that the critical field of the AFM order grows with $B_c \propto J^2$. At the same time, theoretical studies of Kondo insulators suggest that a magnetic field suppresses the Kondo screening, while it enhances transverse spin fluctuations [46-49]. The associated suppression of T_K with increasing magnetic field would shift the critical region, J_c , to higher values of J . Such an intuitive picture

qualitatively agrees with our observations, yet it is clear that a more realistic description is required. In light of the field polarization of the conduction electrons as well as the modification of the crystal electric fields, the implicit assumption of a field-independent J appears oversimplified. Further thermodynamic probes, albeit experimentally challenging, will be required to determine the magnetic structure.”